# The importance of DNA sequence for nucleosome positioning in transcriptional regulation

Malte Sahrhage[1], Niels Benjamin Paul[1,2], Tim Beißbarth[1], Martin Haubrock[1]

**Nucleosome positioning is a key factor for transcriptional regulation. Nucleosomes regulate the dynamic accessibility of chromatin and interact with the transcription machinery at every stage. Influences to steer nucleosome positioning are diverse, and the according importance of the DNA sequence in contrast to active chromatin remodeling has been the subject of long discussion. In this study, we evaluate the functional role of DNA sequence for all major elements along the process of transcription. We developed a random forest classifier based on local DNA structure that assesses the sequence-intrinsic support for nucleosome positioning. On this basis, we created a simple data resource that we applied genome-wide to the human genome. In our comprehensive analysis, we found a special role of DNA in mediating the competition of nucleosomes with cis-regulatory elements, in enabling steady transcription, for positioning of stable nucleosomes in exons, and for repelling nucleosomes during transcription termination. In contrast, we relate these findings to concurrent processes that generate strongly positioned nucleosomes in vivo that are not mediated by sequence, such as energy-dependent remodeling of chromatin.**

## Introduction

The nucleosome is a foundational protein complex in eukaryotic organisms formed by an octamer of pairs of histone subunits (two pairs of H2A/H2B and a tetramer of H3/H4). This complex wraps 147 base pairs (bp) of DNA (1, 2). The positioning of nucleosomes is very important for the regulation of transcription, and the relevance of nucleosome positioning for the regulation of genes can be viewed from many perspectives. The broadest unit to span interaction potential are topologically associated domains (TADs). These are characterized by a loop compartment that brings together potentially interacting proteins at enhancers and promoters of genes close to each other (3). The boundaries of these logical units are set up at so-called insulator regions, which are defined by an interplay of CTCF binding (4) and a consequential strict positioning of a well-positioned nucleosome array (5). Whether the genes inside such units are transcribed is an intricately regulated process involving nucleosome positioning at all levels. The binding of transcription factors (TFs) to promoter and enhancer regions is crucial for the initiation of transcription. As a means of controlling access to transcription factor binding sites (TFBSs), nucleosomes occupy DNA and need active displacement through (pioneer) TFs or energy-dependent remodeling to avoid random triggering of gene expression (6, 7). To achieve an orderly rate of transcription, the RNA polymerase II (RNAPII) is highly regulated by nucleosomes. In particular, the +1 nucleosome directly downstream of the transcription start site (TSS) and a sequential array of well-positioned nucleosomes are needed to reliably initiate transcription (8, 9, 10). Along its way, RNAPII encounters more fuzzily positioned nucleosomes in intronic regions and well-positioned nucleosomes in exons.

Because exonic DNA is the template for the subsequent translation of proteins, the positioning of nucleosomes supports the faultless transcription of these regions by serving as roadblocks to decrease the speed of RNAPII elongation, by marking alternative splice sites (11), and by triggering feedback signals for other concurrent transcription initiation events on the same gene (12).

The transcription ends at the transcription termination site (TTS), which sets up the signal where polyadenylation will be applied to the resulting pre-mRNA. These polyadenylation sites (poly[A] sites) are known to exhibit a considerable nucleosome depletion directly around the DNA-encoded prospective polyadenylation site compared with their nucleosome-occupied surroundings (13).

An extra layer of nucleosome functionality in all of these stages is added through the epigenetic modifications of histone proteins. Depending on the specific biochemical modulations of the histone subunits, nucleosomes can carry marks for the active recruitment of further proteins and show a different flexibility in terms of the energy needed for displacement (14, 15). Because the differential increase or decrease in chromatin accessibility at all of these specific functional sites regulates transcription and thus cell identity, it is crucial to understand the principles that determine nucleosome positioning.

[1]Department of Medical Bioinformatics, University Medical Center, Göttingen, Germany  [2]Department of Cardiology and Pneumology, University Medical Center, Göttingen, Germany

Correspondence: malte.sahrhage@bioinf.med.uni-goettingen.de; martin.haubrock@bioinf.med.uni-goettingen.de

The influences on the positioning of nucleosomes have been debated for many years and can be grouped roughly into DNA sequence, trans-acting factors, and active chromatin remodeling (16, 17, 18). The role of the DNA sequence is particularly controversial. It was postulated that there is a periodic repeat of A/T dinucleotides that support the binding of nucleosomes (19, 20, 21). This pattern has been extended to be represented by local DNA structure (22) with the help of DNAshape (23). The benefits of the supposed patterns for nucleosome support are linked to the flexibility of the DNA strand to facilitate an easier attraction of nucleosomes (24, 25), which can be seen particularly well in extremely stiff poly(dA:dT) tracts that counteract nucleosome binding (26).

Some of these findings have been discovered with the help of a variety of computational tools to predict nucleosome positions, each with a different focus of training data, prediction target, and methodology (27, 28, 29, 30). Among these, there are tools that employ HMMs (31, 32), SVMs (33), and recently more and more convolutional neural network (CNN) architectures (34, 35, 36, 37). The difference in meaning between sequence-intrinsic signals and the actual in vivo positioning of nucleosomes is fundamental to understand. The sequence can only explain the potential of any given position to attract nucleosomes. Whether it is actually occupied by nucleosomes or opened up by regulatory mechanisms cannot be determined without the help of experimental chromatin accessibility data for the particular cell type. In other words, although experimental data can provide insights into the actual nucleosome positioning within cells, it does not directly elucidate the underlying reasons or mechanisms behind such positioning. On the contrary, machine learning on DNA sequence can give an intrinsic estimate of nucleosome attraction potential but cannot answer the question where nucleosomes are located in vivo. Therefore, we need to integrate both information to estimate the overlap of sequence-intrinsic nucleosome support and experimental chromatin accessibility evaluation (38). For that purpose, we use nucleosome occupancy data (MNase-seq) (39), chromatin accessibility data (DNase-seq) (40), and ChIP-seq data for both histone modifications and TFs, mainly in the two cell lines GM12878 and K562 available from ENCODE (41).

In this study, we evaluate the importance of DNA sequence for nucleosome positioning at all elements involved in transcription in the human genome. For that purpose, we developed the nucleosome formation score (NF score), which describes the sequence-intrinsic support for attracting nucleosomes purely based on DNA. The score is derived using a classification algorithm, which is applied in a genome-wide manner. We provide it as a simple data resource, which can be used without further computation for similar analyses in the future. Because of its relative robustness against overfitting, we created a random forest (42) classifier based on the Guo2014 dataset (33) of nucleosomal sequences, which is derived from MNase-seq in human CD4$^+$ T cells (39). Our classifier uses a transformation from raw DNA sequences to a frequency spectrum of local DNAshape (propeller twist, helix twist, minor groove width, electrostatic potential, obtained with the DNAshapeR package (43)). This binary classifier was applied in a sliding-window

manner to the human genome in two different resolutions, yielding a genome-wide mapping of sequence-intrinsic nucleosome support.

We aspire to give a comprehensive overview of the crucial parts of the transcriptional machinery and provide our genome-wide measure for sequence-intrinsic nucleosome support as a simple data resource.

Our findings are embedded into a framework of previous studies that showed the relevance of DNA for nucleosome positioning for some specific elements, and we add some new insights about the sequence definition of nucleosome positioning in transcription. We demonstrate that there is a clear influence of DNA sequence for supporting nucleosome binding at positions of high competition with other DNA-binding proteins and where it is beneficial to accumulate nucleosomes by default, such as exons, or repel them in TTSs. In contrast, we give examples of positioned, but sequence-independent, nucleosome arrays in proximity to the TSS or around insulator sites and put our findings into the context of more unspecific, fuzzy nucleosomes.

## Results

### Sequence-intrinsic nucleosome support characterizes functional genomic regions

We constructed a resource to estimate the support of DNA sequence for attracting nucleosomes based on a random forest classifier trained on nucleosome occupancy data. This nucleosome formation (NF) score was applied to a range of different genome locations. A broad range of such entities can be found in the histone ChIP-seq–derived mappings of chromHMM for the cell lines K562 and GM12878 (44). Fig 1A shows the mean sequence-intrinsic nucleosome support (NF), the nucleosome occupancy (MNase-seq), and the chromatin accessibility (DNase-seq) for the 147 bp in the center of any given chromHMM fragment. In total, there are 622,257 fragments in K562 and 571,339 in GM12878. A list of the individual fragment numbers for each subgroup can be found in Table S1. We grouped all chromHMM states into the three categories: "regulatory" (promoters/enhancers), "heterochromatin/repetitive," and "transcription" (transcriptional transition/elongation, etc. and insulators).

The most intriguing feature is the relation between sequence-intrinsic nucleosome support and actual nucleosome occupancy. Active promoters show the highest DNA-intrinsic nucleosome support (NF: ~0.8). However, being crucial places for transcriptional regulation, they are also the most accessible regions (low MNase-seq of 0.4–0.55 and high DNase-seq of 0.65–1). The same nucleosome support can be observed in poised promoters, but with closed chromatin (low DNase-seq, high MNase-seq). This observation can be put in contrast to the highlighted group of heterochromatin fragments. Here, we observe rather low nucleosome support in both cell lines of roughly 0.4. At the same time, the locations are very inaccessible with a nucleosome occupancy of 1 and a very low DNase-seq signal of 0.03.

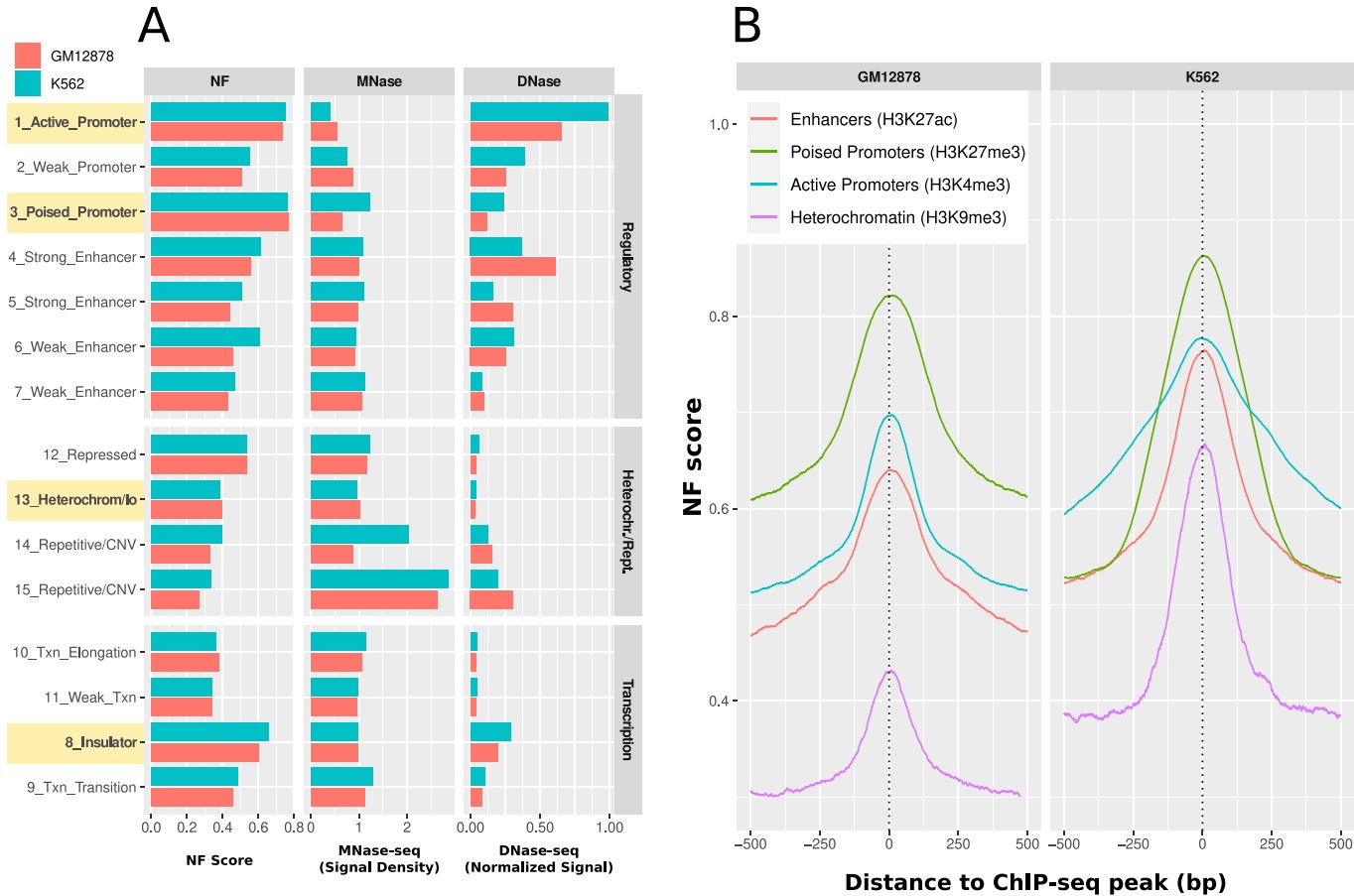

**Figure 1. Nucleosome support in different regions of the genome and around histone modification ChIP-seq peaks.**
**(A)** chromHMM. Nucleosome support (NF score), nucleosome occupancy (MNase-seq), and chromatin accessibility (DNase-seq) in all states of the chromHMM mappings. The highlighted regions exhibit a high NF score in active regulatory regions (active and poised promoters) and in insulators, as opposed to low nucleosome support regions in heterochromatin. Active promoters are rather accessible, and poised promoters are closed off mostly. Heterochromatin exhibits pronounced inaccessibility, whereas insulators demonstrate hybrid accessibility characterized by relatively elevated levels of both DNase-seq and MNase-seq signals.
**(B)** Nucleosome support of histone modifications. A distinct peak is observed in proximity to each histone modification ChIP-seq peak, providing evidence of the influence of DNA sequence on the positioning of individual nucleosomes. This influence is higher in marks indicating a regulatory function such as poised promoters (H3K27me3, green), active promoters (H3K4me3, blue), and more generally accessible chromatin (H3K27ac, red) as compared to repressing histone marks (H3K9me3, purple).

The last highlighted group in Fig 1A are insulators. In this group, we can observe a relatively high nucleosome support of 0.6–0.66 and a medium accessibility with relatively high values for both MNase-seq of 1 and DNase-seq signal of 0.2–0.3.

As a side note, it can be observed that enhancers show the most variability in their status. This effect on the NF score is most prominent in the groups of Weak_Enhancer (groups 6 and 7), in which the NF score varies almost about 0.2 between the minimum in GM12878 and the maximum in K562. These groups are the most ambiguously defined ones, however, based on their histone marks (44). Nevertheless, also in the Strong_Enhancer group (group 4) and the Active_Promoter group (group 1), there is a comparatively high difference in chromatin accessibility. This could be due to experimental biases but because the direction of the differences is not the same in both instances.

In summary, these observations suggest that regions involved in active regulation of transcription have significantly higher levels of nucleosome support with well-defined sequence context. Whether these important locations are actually occupied by nucleosomes is

dependent on whether they are used in vivo (active versus poised promoters). By comparison, constitutive heterochromatin has a much lower sequence-intrinsic nucleosome support while being mainly inaccessible.

Histones can carry different kinds of modifications that determine their function and the flexibility of the surrounding chromatin accessibility (14, 15). Because the chromHMM mapping is derived from ChIP-seq data of histone modifications, we can gain more detailed insights into the local nucleosome support relative to the actual position of the histone modification. Therefore, we analyzed 518,609 histone ChIP-seq peaks from ENCODE for the GM12878 and K562 cell lines. We chose the histone marks according to their role in relation to the derived chromHMM states. The H3K4me3 peaks are representative of active promoter histone marks (GM12878: 194,130; K562: 240,828), H3K27ac is representative of enhancers (GM12878: 114,738; K562: 102,352), H3K27me3 represents poised promoters (GM12878: 100,266; K562: 477,102), and H3K9me3 is repressive chromatin as a general indication for constitutive heterochromatin (GM12878: 37,756; K562: 11,778). Whenever there are

multiple equivalent experiments for the same target, we calculated the mean over all experiments in a single line.

Fig 1B shows the average NF score around the set of different histone modification ChIP-seq peaks. It can be seen that, in the specific experimental configurations, the previous findings of a higher NF score in regulatory important regions can be confirmed. For both cell lines, it can be observed that the three activating marks are higher than the heterochromatin mark. In addition, all histone modifications show a clear peak of nucleosome support around the central ChIP-seq point-source peak. The poised promoter mark H3K27me3 is consistently the most sequence-positioned in both cell lines and can be found at NF score levels of 0.82 and 0.86, respectively. The next highest pattern on both sides is the active promoter mark H3K4me3, which shows a peak at an NF score of 0.67–0.78. The general accessibility mark in promoters and enhancers H3K27ac is next at 0.64–0.76, and the peak of the heterochromatin mark H3K9me3 is located at the lowest NF score level of 0.44–0.66. These results indicate a local sequence preference for nucleosome binding around the central peak position of the histone modifications with a distinctly higher nucleosome support for regulatory elements. Because the modified histones are part of the nucleosome, it can indeed be expected that the NF score is highest around its central binding places, but it is remarkable that the sequence-intrinsic influence is highest at putative positions of interaction between nucleosomes and proteins such as TFs.

## Regulatory nucleosomes: competition between sequence-intrinsic nucleosomes and transcription factors

### Promoter regions

The previous analyses suggest that DNA structure plays different roles in nucleosome binding in different biological environments. To enhance our comprehension of these distinctions, we direct our attention toward regulatory loci, specifically encompassing all human promoters. This analytical approach allows us to establish correlations between the unchanging attributes of sequence-intrinsic regulatory potential and its consequential influence on gene regulation. The promoters are defined as 1 kb upstream of all 30,142 RefSeq-defined human genes (see the Materials and Methods section). Alternative TSS locations are counted as individual promoters, and strand direction is aligned to 5′-3′ direction.

We applied the high-resolution NF score (see the Materials and Methods section) to all human promoters and compared this cell-type unspecific signal with the actual experimental evidence of chromatin accessibility in K562 and GM12878, measured by ENCODE MNase-seq, DNase-seq, and H3K4me3 ChIP-seq data, as well as a derived DHS score, which integrates DNase-seq peaks over 403 primary cell lines (see the Materials and Methods section). In addition, we include PhyloP, a bp resolution evolutionary conservation score.

Fig 2A shows the position-specific mean distribution for all available genes in relation to the TSS. An extended version of this figure showing 1 kb up- and downstream of the TSS can be found in Fig S1. The overall trend of the different data sources is very similar between both cell lines and confirms the general expectations of chromatin accessibility. Although the general occupation with nucleosomes declines toward the TSS (MNase-seq from 0.9 to 0.5), the amount of promoter histone modification H3K4me3 increases toward the TSS (from 10 to 25 in K562, from 4 to 7 in GM12878) with a distinct valley about 100 bp upstream of the TSS. This decline in nucleosome occupancy can be explained simultaneously with the gain of chromatin accessibility by putative proteins such as TFs, which is described by a peaked rise of DNase-seq (from 0.25/0.3 to 1.0/1.6) and a DHS score (from 10 to 145) in the same area. The NF score increases toward the TSS with a more steady, less peaked slope from ~0.5 at −1 kb to 0.85 at the TSS. The evolutionary conservation (PhyloP) stays relatively stable at a level of 0.09 until it increases steeply to 0.45, starting at −200 bp and continuing with a high evolutionary conservation downstream into the gene.

The average profiles of each signal in Fig 2A are generally coherent, primarily illustrating the concentration of DNA accessibility in proximity to the TSS within the nucleosome-depleted region. However, to understand the local interaction between nucleosomes and TFs, it is crucial to examine the specific positions within the entire promoter region. Fig 2B shows examples of individual promoters and the relations between nucleosome binding, chromatin accessibility, and the potential nucleosome support described by the NF score.

The transcription factor NANOG is a general developmental factor involved in the proliferation and renewal of embryonic stem cells (46), and thus, it does not play a role in the differentiated cell lines. Fig 2B (left) displays the integrated profiles for the NANOG gene, which is not expected to be actively transcribed in either cell line. Therefore, the first significant observation is the lack of clear signals in both DNase-seq and H3K4me3 ChIP-seq data. In addition, the general accessibility in primary cell lines, described by the DHS score, is mainly zero overall, too, albeit the NF score is high near the TSS (highlighted region) with an NF score of 0.9–0.95. At the same location, a maximum of nucleosome occupancy (MNase-seq of 4.4 in K562 and 3.6 in GM12878) can be observed. The evolutionary conservation at this point is much higher with up to 2.9 compared with the near-zero level in the rest of the promoter. This indicates a regulatory nucleosome position that is currently blocked by a well-positioned nucleosome so that the activation of the NANOG gene requires an active displacement of this nucleosome.

In contrast, the transcription factor TP53 carries substantial importance in both the K562 and GM12878 cell lines, as it functions as a critical tumor suppressor (47). Thus, this gene stands as a representative case of a promoter subject to positive regulation. The profiles for this TF can be found in Fig 2B (right). The accessible regions are separated into two distant peaks on the level of DNase-seq and DHS score, one of which is located around the TSS and the other one at 800 bp upstream. Both accessible regions seem to be important in almost all primary cell lines (DHS score of 401 at the upstream peak and 398 downstream).

Both of these peak areas furthermore exhibit an absence of nucleosomes in these areas, evident in an MNase-seq signal of 0–1 at the upstream peak and of 0–0.3 in the downstream peak. Interestingly, the downstream peak is flanked by a high nucleosome occupancy of up to 1.5–2.1 to either side, potentially marking shifted nucleosomes and/or the well-described +1 and −1 nucleosome downstream of the TSS. The sequence support for nucleosome binding in the form of an NF score shows a broader plateau of 1 at

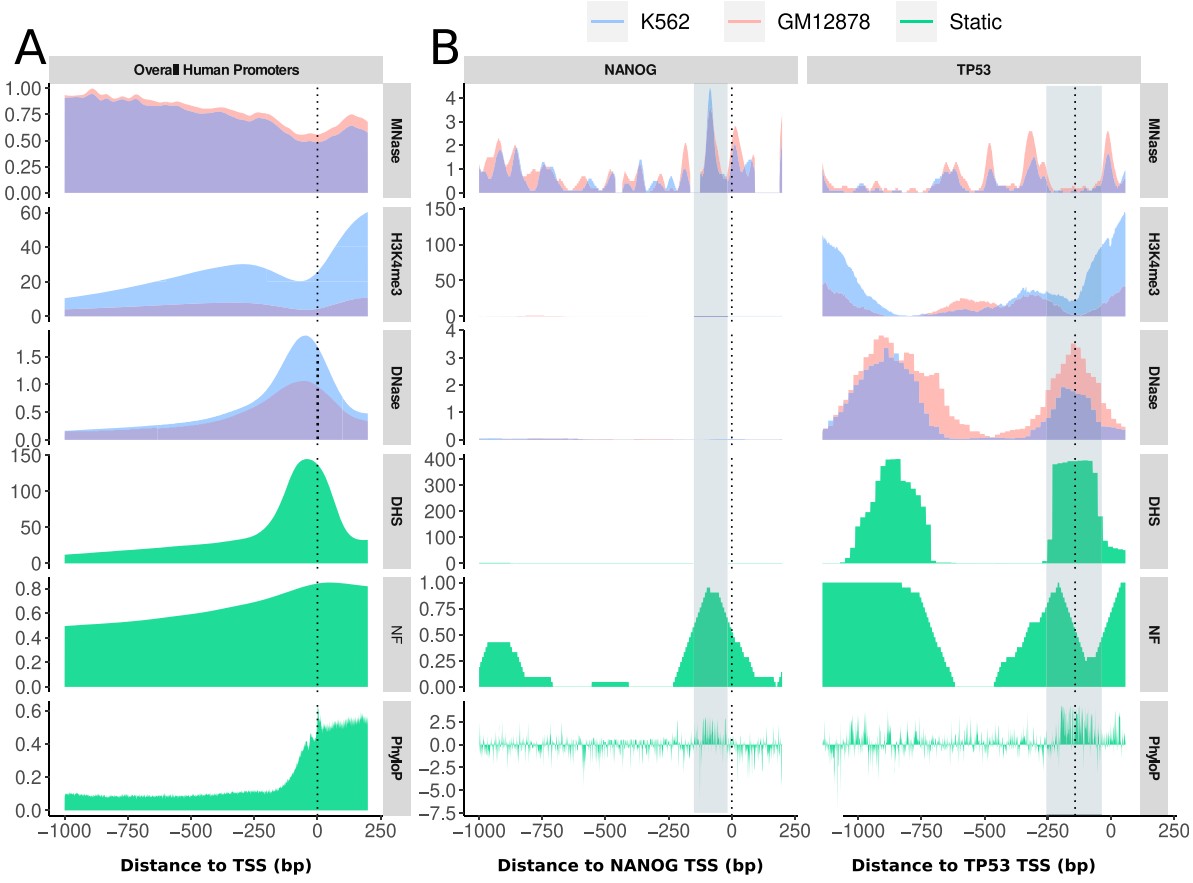

**Figure 2. Chromatin accessibility of human promoters.**
**(A, B)** Profiles for experimental chromatin accessibility (MNase-seq, DNase-seq, H3K4me3 ChIP-seq) and static data (green: DHS score, NF score, PhyloP) as (A) the mean of all human RefSeq promoters and (B) the promoters of the individual genes *NANOG* and *TP53* in K562 (blue) and *GM12878* (red). The dotted line at position 0 marks the TSS. **(A)** Overall promoter regions. The chromatin accessibility increases, whereas the general nucleosome occupancy decreases. The sequence-intrinsic nucleosome support (NF score) increases toward the TSS along the increasing chromatin accessibility. Furthermore, the H3K4me3 ChIP-seq and MNase-seq signals exhibit an NDR upstream of the TSS, and the evolutionary sequence conservation represented as PhyloP score is generally decreased in the promoter region but increases steeply toward the TSS. **(B)** *NANOG profiles.* In the context of the two cell lines, NANOG, because of its role as a developmental pioneer factor, does not exert any influence in K562 and GM12878. The marked area denotes an overlap of high sequence-intrinsic nucleosome support and a high MNase-seq peak. That means the main site of nucleosome attraction potential is occupied by well-positioned nucleosomes. *TP53 profiles.* TP53 is an important tumor suppressor in both cell lines. The NF score follows the peaks of high accessibility in the DHS score and DNase-seq in location. In the annotated instance, the region characterized by elevated nucleosome support exhibits a nucleosome-free state in vivo, indicating a displaced nucleosome position. Both gene promoters show an increased PhyloP score at the high NF score sites, indicating an evolutionary conservation of the nucleosome-supporting sequence. Overall, it should be noted that the NF score correlates positively with the accessible chromatin data (DHS, DNase-seq) in actively used promoters but correlates with nucleosome positioning data (MNase-seq) in closed promoters, demonstrating the original nucleosome positioning potential by the DNA sequence, which is overwritten upon promoter activation. y-Axes: MNase—density graph of signal enrichment; H3K4me3—control normalized tag density; DNase—read depth normalized signal; DHS—DHS score (see the Materials and Methods section); NF—NF score (high resolution); PhyloP—PhyloP score.

the upstream portion of the accessible promoter and another maximum directly upstream of the TSS in the nucleosome-depleted region with a sharper profile and a short decline before it peaks again closely downstream of the TSS.

Fig 2A and B together leads us to say that locations in promoters where TFs compete with nucleosomes directly are the places with the highest nucleosome support. Whether these are occupied by nucleosomes is dependent on whether the gene is actually expressed. Thus, we are not of the opinion that all nucleosomes in the promoter are sequence-positioned, as suggested in the mean of all promoters in Fig 2A. In these overall promoter regions, the NF score stays constantly relatively high, but in the individual promoters in Fig 2B, it depends on the co-

localization with competing TFs that mediate chromatin accessibility at local, defined places. The NF score exhibits a positive correlation with nucleosome occupancy in inactivated genes at regulatory key nucleosomes, where the nucleosome is deliberately positioned by sequence. In contrast, it correlates negatively once the corresponding nucleosome is displaced by targeted TF action at the same location.

### Transcription factors
The regulation of gene expression is highly dependent on the binding of TFs. TFs exhibit a range of strategies in their competition with nucleosomes, displaying remarkable adaptability. Nevertheless, a common feature among them is that a region occupied by

TFs should typically be devoid of nucleosomes. To give an overview of the nucleosome support for the diverse array of TFs, we calculated the mean NF score for all available ENCODE TFs at the peak positions of optimal IDR threshold peaks. The number of available files with at least two biological replicates is 151 for GM12878 and 391 for K562. Fig 3A shows a histogram of NF scores at each according individual peak point position as defined in the Methods section. It is notable that the overall mean nucleosome support of TFs is in general rather high with ~0.71 and very consistent between the two cell lines with a difference within less than 1%. The difference in appearance between these two groups lies solely in the imbalance in the number of available experiments for each cell line. Furthermore, we calculated the mean NF score around the TF ChIP-seq peak location summarizing all of the TFs into one profile, which is depicted in Fig 3B. The peak of the NF score coincides precisely with the center of the experimentally validated ChIP-seq peaks. As expected, the NF score at this position matches precisely with the median NF score at the peak position depicted in Fig 3A's histogram, measuring 0.71 at the peak's maximum position. The figure illustrates that, in the vicinity of TF binding events, the MNase-seq signal experiences a consistent reduction (GM12878: signal decrease from 0.91 to 0.85; K562: signal decrease from 1.01 to 0.89). In contrast, the NF score reaches its peak precisely at these locations, suggesting the presence of sequence-intrinsic nucleosome positioning, particularly in regulatory regions that compete with initially well-positioned nucleosomes. The NF score lies in between what is to be expected for enhancer and promoter regions considering Fig 3. Because of the abundance of ChIP-seq peaks that are expected to be located in enhancer regions, the profiles might be biased toward enhancer TFs. To show whether promoters and enhancers behave differently, we added the information of H3K4me3 and H3K27ac signal for the TF ChIP-seq peaks in Fig S2. In this context, it is apparent that both signals exhibit a consistent depletion pattern in close proximity to the binding locations, demonstrating a strong similarity in their behavior.

Because ChIP-seq only covers the final state of the displacement of the nucleosomes, we illustrate the relation of originally closed off chromatin and the sequence-based nucleosome positioning with a pioneer factor that serves the purpose of displacing the nucleosome and making the chromatin accessible. The pioneer transcription factor GATA3 has been described to bind well-positioned nucleosomes (48). To check whether this positioning can be seen on the DNA level as well, we analyzed 43,504 GATA3 binding sites with our global NF score and compared them with published nucleosome occupancy (MNase-seq) in MDA-MB-23 cells (49). Fig 4A shows the profiles of +500 bp around GATA3 binding sites, once in cells with GATA3 expression (+GATA3) and once with GATA3 knockout (−GATA3). It is evident that the area is supportive for nucleosome binding based on sequence as shown by an NF score that shows an increased peak of ~0.15 around the binding sites from 0.35 to 0.5. In the case of cells without GATA3 expression, there is a shift of increased nucleosome occupancy (0.63–0.7) around the sites, showing increased nucleosome occupancy on top of the closed binding sites. However, upon GATA3 expression, the peak observed in the data

undergoes a split into a double peak at 0.675, corresponding to both sites, while exhibiting a distinct valley at 0.62 directly at the TFBSs. This behavior suggests the presence of a robust nucleosome that was initially positioned based on the DNA sequence but subsequently displaced because of the influence of pioneer transcription factors.

### Insulators

Another category of regions with a rather high mean NF score of 0.66 in Fig 1A were insulator regions. Insulators are DNA regulatory elements that play a crucial role in organizing and maintaining the spatial organization of the genome. The protein CTCF is a key factor in mediating insulator function by facilitating chromatin looping and regulating gene expression (50). The nucleosome occupancy and sequence-intrinsic nucleosome support around 43,247 ChIP-seq peaks for CTCF in K562 and 43,631 peaks in GM12878 are compared in detail in Fig 4B.

The nucleosome occupancy shows a very stable, phased array of nucleosomes around the CTCF peaks with an ~200-bp periodicity and an ongoing MNase-seq signal decrease per peak from 1.8 to 1.2 with increasing distance to the TSS. Although the nucleosome occupancy is lowest directly at the CTCF peaks, the sequence-intrinsic nucleosome support is rather high specifically in these regions with an NF score of about 0.77 at the peak. The observed periodicity in nucleosome occupancy is completely absent on the sequence level, indicating a strong influence on nucleosome positioning at CTCF sites but active energy-driven chromatin remodeling once the site was cleared and the factor is bound to the DNA.

## Sequence-intrinsic nucleosome support in transcriptional regulation

In the previous part, we focused mainly on the role of DNA for nucleosome positioning in promoters, in proximity to the TSS and in competition with TFs. The following sections will explore other functional elements of the genome along the transcriptional axis in a similar way.

### Active chromatin remodeling upon transcription initiation

It is known that there are a multitude of influences on the dynamic positioning of nucleosomes. One such phenomenon is the phasing of ordered nucleosome arrays downstream of the TSS upon transcription initiation (51). Fig 5A shows the comparison of nucleosome occupancy (MNase-seq) and the DNA-intrinsic nucleosome support (NF) around the TSS of all human RefSeq genes. The proposed periodic downstream array of nucleosome positions is apparent, marked by arrows, stretching with a periodicity of 200 bp at MNase-seq levels between 0.75 and 0.9 (in GM12878) or 0.65 and 0.85 (in K562), respectively. At the same time, although the NF score is generally still relatively high close to the TSS and the +1 nucleosome with a value of 0.8–0.85, the periodicity is not mirrored on the sequence level and the general sequence-intrinsic nucleosome support decreases further downstream. Therefore, it is reasonable to conclude that the dynamic relocalization of nucleosomes because of ATP-

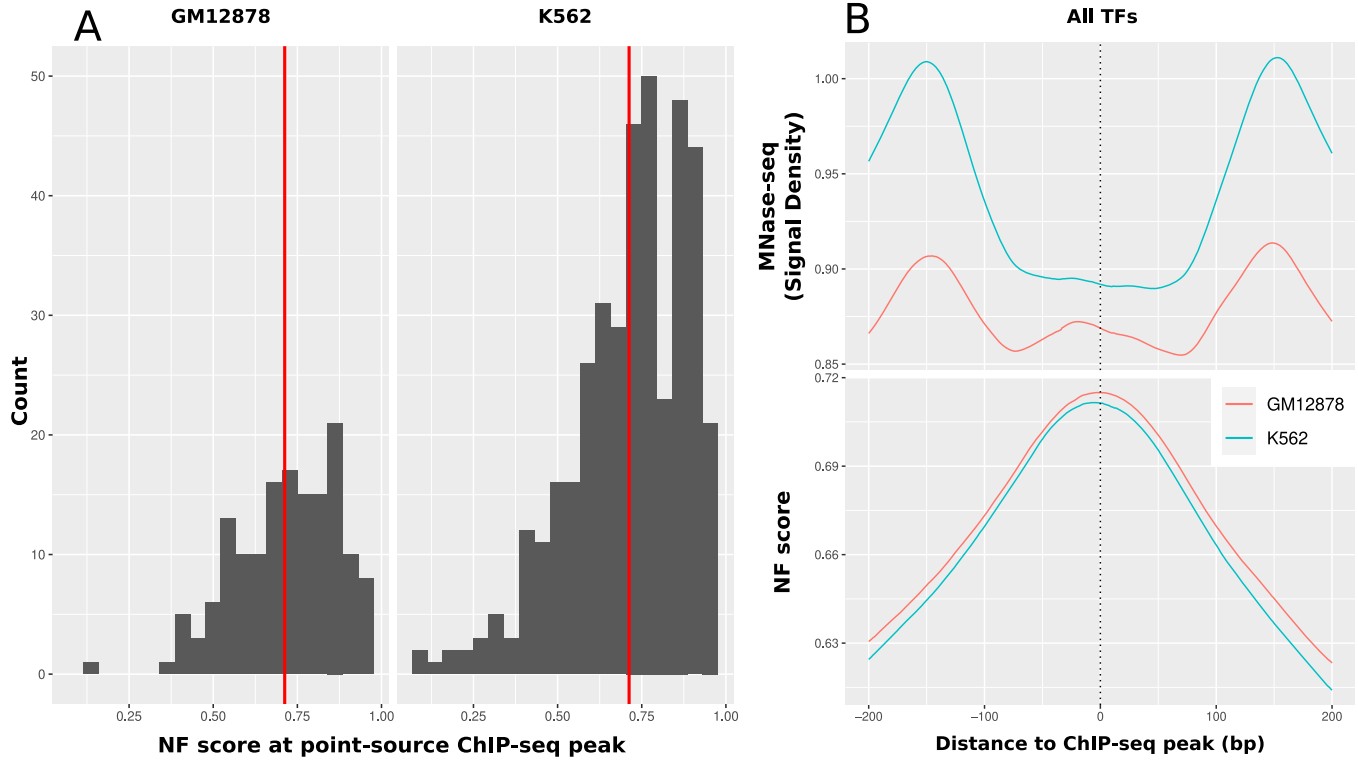

**Figure 3.** **Overall sequence-intrinsic nucleosome support around human TF binding locations.**
**(A)** Sequence support at TF binding locations. Histogram of the NF score at the point-source peak position of all available ChIP-seq peaks on ENCODE for K562 and GM2878. The red line marks the mean of the NF score at peaks (GM12878: 0.7150, K562: 0.7114). **(B)** Average sequence-intrinsic nucleosome support (NF score) around the point-source peak of all TFs in comparison with nucleosome occupation (MNase-seq) in K562 and GM12878. It can clearly be seen that the sequence-based positioning signal is highest around the ChIP-seq peak location (0.71), whereas this is the position with the lowest nucleosome occupancy in vivo. This is evidence for the high importance of sequence to position nucleosomes based on the sequence at places where competition with TFs is highest.

dependent remodeling can overwrite the sequence-mediated nucleosome guidance.

### Sequence-intrinsic nucleosome occupation potential of exon regions

Although it was just shown that there is a sequence-independent creation of nucleosome arrays downstream of the TSS, that does not entirely describe the sequence-intrinsic local nucleosome support for other regions along transcriptional regulation. It is described in literature that exons are targets of a particularly high nucleosome occupancy, which is related to transcription speed of RNAPII and the recognition of alternative splicing (11). To test whether this principle is also evident on the nucleotide sequence level, we analyzed the exons of all human genes by comparing the MNase-seq signal for K562 and GM12878 and the NF score again. Fig 5B shows both average signals centered around 267,864 exons with strand orientation taken into account. The figure on the left employs an alignment centered around the intron/exon (IE) junction, specifically focusing on the exon starting points. Conversely, the right-hand figure emphasizes the alignment based on the central exon midpoints. Generally, the figures indicate that there is a nucleosome positioned over the center of the exon. The midpoint-centered figure shows that the nucleosome occupancy and the NF score show a distinct peak over the exon regions. The extent of the

rise of the NF score toward the exon center is as high as ~20% from slightly above 50% to almost 70% and is distributed mainly between 100 bp upstream and 150 downstream of the exon midpoints. The nucleosome occupancy mirrors this pattern of a central peak of about 0.3–0.4 from 0.9 up to 1.25 in both cell lines around the exon. In addition, a notable observation indicates a relative depletion of MNase-seq signal directly upstream of the peak, which is more clear when aligning the exons over the IE junction. In this view, it is also apparent that the main nucleosome is positioned directly downstream of the IE junction. That the middle-centered nucleosome can be understood as equivalent to the nucleosome directly downstream of the exon start can be explained with the median of the RefSeq exon length being 129, thus being not large enough to fit multiple nucleosomes. The general tendency of a single sequence-positioned nucleosome in the center of an exon holds true when separating the exons by length. For a more detailed analysis of these patterns, Figs S3 and S4 provide an enhanced representation by separating the exons based on their length, again around the midpoints and around the IE junction, respectively. The images shown here encompass all human exons. Because the first exon is essentially the TSS, Fig S5 shows the same profiles, once for all exons except the first of each gene and once only for the first exons. Because of their comparatively small amount, the principal image does not change when filtering out the first exons. The isolated first

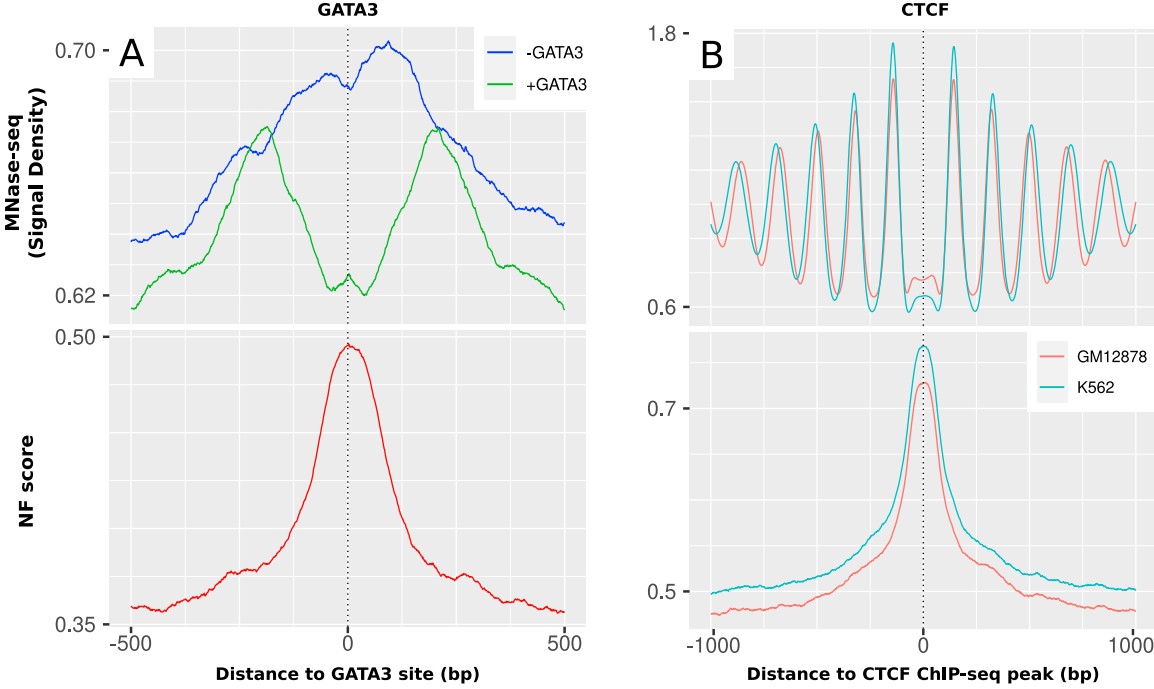

**Figure 4. Nucleosome formation (NF) score and nucleosome occupancy (MNase-seq) surrounding the transcription factors GATA3 and CTCF.**
**(A)** GATA3. GATA3 binding sites of MDA-MB-23 cells with (green) and without (blue) GATA3 expression. The nucleosome occupancy is higher toward the binding sites in the cells without GATA3 expression. This peak is depleted in the GATA3 expression cells, and a double peak of nucleosome occupancy is left around the sites with a distance of ~200 bp. The nucleosome support displays a pronounced peak precisely overlapping the TFBSs, indicating the presence of an intrinsically well-positioned nucleosome that is actively displaced by GATA3. **(B)** CTCF. CTCF ChIP-seq peaks from ENCODE in GM12878 and K562. The nucleosome occupancy shows a very distinct periodicity around CTCF peaks. This periodicity of nucleosome positions is not apparent on the level of sequence-intrinsic nucleosome support, which exhibits a single peak directly around the CTCF peak.

exons, however, do not exhibit the same occupation of exons but rather show the same nucleosome positioning as shown in the TSS analysis in Fig 5A with the clear +1 nucleosome.

### Sequence-dependent nucleosome occupancy prevention around poly(A) sites in transcription termination

Another important part of gene regulation and splicing is the nucleosome occupancy at poly(A) sites as the place for transcription termination. These were shown to be explicitly free of nucleosomes, thus making them accessible for proteins marking the sites and interacting with RNAPII in transcriptional termination (13). We obtained the ENCODE MNase-seq signal for K562 and GM12878, as well as the NF score, for a list of 570,740 poly(A) sites in the human genome (52). These are classified by the original authors into clusters of biological origin. All sites are oriented by strand in 5'-3' direction.

Fig 5C shows the nucleosome occupancy and NF score for all poly(A) sites in the mapping together. There appears to be a direct positive relation between the sequence-intrinsic nucleosome support and the nucleosome occupancy. Both nucleosome occupancy and sequence-intrinsic nucleosome support exhibit a decrease around the sites in the central 250 bp. There is a 20% drop in nucleosome support relative to the already low level of 35%. It can be noted that the sequence support for nucleosome binding seems to be slightly higher in the upstream direction. The same overall decrease is apparent on nucleosome occupancy in both cell lines with a drop of 0.25–0.3 in MNase-seq signal. A proposed increase in nucleosome occupancy downstream of the poly(A) site in vivo

relative to the upstream direction (13) cannot be observed overall. Nevertheless, outcomes for specific subgroups of poly(A) sites can differ. Fig S6 gives an overview over all poly(A) sites separated by clusters defined in the PolyASite 2.0 atlas. There, it can be observed that the general repelling trend can be observed for all subclusters, except for sites that coincide with exon regions, which exhibit a similar pattern as shown above for exons, that is, a sequence-positioned nucleosome over the exon. The aforementioned difference of a higher nucleosome occupancy downstream of the site can be shown there for the DS cluster, which denotes the TTS 1 kb downstream of a terminal exon. In contrast, the higher NF score in the mean profile upstream of the sites can most probably be explained by the group of sites located in the AU group, in which the signal is higher because of its proximity to the TSS, which has been demonstrated to have a rather high NF score in Fig 5A.

### Low-frequency differences in HelT are the most discriminative sequence feature for nucleosome positioning

The random forest was selected for modeling the classifier underlying the NF score because of its intrinsic ability to generate feature importance scores. The input features consist of the power spectral density (PSD) of frequencies inherent in underlying DNAshape structures (see the Classifier section in the Materials and Methods section). In Table 1, the top 10 input features for distinguishing between nucleosomal and linker DNA are listed. The mean decrease in Gini impurity is given for all frequencies of the PSD-transformed DNAshape

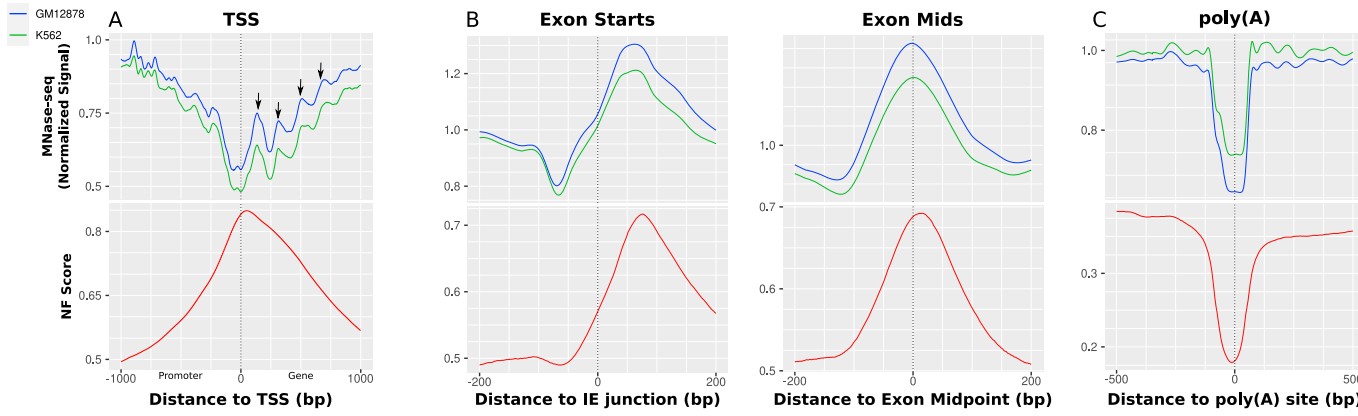

**Figure 5. MNase-seq and NF score for different elements from transcription initiation to transcription termination.**
**(A)** Promoters. The profiles are consistently aligned to ensure that the promoter located at the 5′ end is consistently positioned on the left side. The central vertical line represents the TSS location. The nucleosome occupancy declines around the TSS and shows an apparent periodicity in both directions. The NF score rises in anti-correlation toward the TSS and does not indicate the proposed periodicity. This implies there is support for nucleosome binding for the NDR, which is accessed by TFs but no sequence support that places the periodically following nucleosomes in the downstream array. **(B)** Exons. The left image is centered specifically on the intron/exon (IE) junction, whereas the image on the right aligns the exons precisely over their midpoints. The NF score shows high sequence-intrinsic nucleosome support and an experimentally high nucleosome occupancy directly downstream of the exon starts, which can be observed even more stably over exon centers. This correlation strongly suggests the functional significance of the observed sequence-defined nucleosome. **(C)** Poly(A). There is a decline in both sequence-intrinsic nucleosome support (NF score) and nucleosome occupancy (MNase-seq) around the transcription termination sites. This behavior indicates nucleosome-repelling positions, encoded in DNA sequence.

values. It is evident that most of the best input features to distinguish between nucleosomal and linker DNA are low-frequency differences in HelT. In the original DNA structure space, low frequencies indicate long-range structural changes rather than short local perturbations.

As HelT denotes the rotational angle between two adjacent bp along the helix direction, this effect can be understood as what happens when taking both ends of a 147-bp DNA double strand and twisting them in opposite directions. Fig S7 shows the visualization of the mean profiles for each DNAshape feature, as well as the PSD-transformed information for both nucleosomal and linker DNA from the training set, and underlines the differences between both groups in the low-frequency structure changes visually. It has been shown in this study that the NF score is highest at regulatory meaningful positions, which are targeted by TFs. It is worth noting that the long-range structural disparities between nucleosomal and linker DNA contrast with the localized and short DNA structure features known to be recognized by TFs at their binding sites (53). Therefore, attraction for the binding of nucleosomes and for TFs is encoded at the same locations but on different frequency ranges.

## Discussion

In this study, we analyzed the role of DNA sequence for the positioning of nucleosomes. For that purpose, we developed a random forest classifier to distinguish nucleosomal and linker DNA and used it to derive a score to estimate the sequence-intrinsic nucleosome support of different functional elements in the human genome. The degree to which nucleosome positioning relies on DNA sequence has been the subject of extensive debate (16). These questions cannot be answered by analyzing either chromatin accessibility experiments or DNA sequence individually. On the one hand, the experimental verification of chromatin accessibility for

any given cell type will only describe the result of the multitude of counteracting influences on nucleosome positions. On the other hand, DNA can explain the importance of nucleotide sequence on their potentially favored binding locations, without reliably predicting the actual nucleosome occupancy in any given cell type in vivo. Therefore, we compared the results of our sequence-intrinsic nucleosome support in the form of the NF score with experimental data for chromatin accessibility and histone modifications and described the possible combinations of in vivo nucleosome positioning with its underlying DNA sequence support. We could demonstrate that DNA-intrinsic nucleosome support is not directly equivalent to the resulting nucleosome occupancy in a specific cell line in vivo. Rather, there is a positive influence on nucleosome binding in the DNA sequence of genomic regions, which are involved in actively regulating transcriptional processes, such as promoters, enhancers, exons, and in the center of insulators. Simultaneously, certain locations exhibit lower nucleosome support, which can serve functional purposes, such as well-positioned nucleosome occupancy arrays downstream of the TSS and in the vicinity of insulators. Conversely, there are instances where lower nucleosome support indicates non-functionality, as observed in constitutive heterochromatin. That sequence-positioned nucleosomes in regulatory regions are a functional effect and not the result of experimental biases in relation to linker sequences in the underlying training data from Guo et al is illustrated by Fig S8. This image compares the training sequences in terms of their relative proportion over different biological contexts. There are no major differences in the origin of both categories, which are mainly distributed over the most abundant regions of the human genome such as distal intergenic regions and introns.

The increased reliance on DNA sequence in gene regulatory regions could be shown in this study from a variety of biological angles. The apparent general trend in promoter regions could be

**Table 1. Feature importance of the top input features for the random forest classifier as a mean decrease in Gini impurity.**

|  | Feature | Freq 1/147 bp | Gini importance |
|---|---|---|---|
| 1 | HelT | 5 | 62.16 |
| 2 | HelT | 1 | 60.33 |
| 3 | HelT | 2 | 55.22 |
| 4 | HelT | 4 | 52.84 |
| 5 | HelT | 3 | 47.88 |
| 6 | HelT | 6 | 40.64 |
| 7 | ProT | 4 | 34.45 |
| 8 | ProT | 3 | 27.75 |
| 9 | ProT | 5 | 25.33 |
| 10 | ProT | 8 | 23.84 |

It can be seen that the top input features consist of low-frequency signals of HelT.

narrowed down to be due to the direct places of competition with TFs. The nucleosome support by sequence could be attributed to evolutionary pressures favoring the development of DNA sequences that facilitate optimal nucleosome binding, effectively regulating access to these regions. This effect has been described to be particularly important for organisms of higher complexity (55). Especially, these highly sequence-intrinsic and evolutionary conserved positions can make a large difference in the regulation of gene activity, and thus, the sequence can help to increase the resistance to accidentally triggering gene expression. The counteracting process to this state is an active eviction of the blocking nucleosome by a combination of (pioneer) TFs, histone modification enzymes, and active chromatin remodeling to gain accessibility for a particular position (56). Because of the inclusion of the relevant TFBSs within the confined nucleosome positions, it can be inferred that there are a presumed co-evolution of proficient nucleosome organization and the existence of high-quality binding sites for activating TFs. Therefore, we interpret the crucial function of DNA in these "regulatory" nucleosomes as a mechanism to regulate the competition between nucleosomes and TFs for spatial availability. This competition affects the energy required to activate biological processes, allowing for dynamic control and modulation. This co-localized competition raises the question of how sequence-encoded nucleosome preferences and well-defined TFBSs can evolve at the same positions. The answer lies in the analysis of feature importance derived from the random forest model. This study demonstrates that the primary predictors for distinguishing between nucleosomal and linker DNA stem from low-frequency, that is, long-range, differences in HelT. Simultaneously, it is well documented that TFs recognize very local, high-resolution, that is, high-frequency, DNA structure signals (53). This leads to the assumption that the nucleosome attraction and the counteracting TFs are encoded at the same positions but on different frequencies of DNA structure changes. Further evidence for the systematic co-localization of regulatory proteins (high DHS score) with highly nucleosome-attracting regions (high NF score) can be observed in Fig S9. That the accessibility of these sequence-defined nucleosomes plays a significant role in regulating gene expression is

further supported by Fig S10, in which gene expression values are compared between groups of genes based on a correlation or anti-correlation of the NF score and chromatin accessibility data. The role of DNA sequence for different kinds of TFs under this assumption has been partially analyzed for different TF families (57, 58). Nevertheless, it would be worthwhile to research the differences in sequence-intrinsic nucleosome support in regard to pioneer functionality. The co-evolution to occupy highly important places would suggest a high nucleosome support at pioneer TFBSs. However, this presents a notable challenge as the binding sites of pioneer transcription factors are often located in close proximity to the subsequent TFs that bind in a consequent step and that are not directly in competition with the nucleosome. Furthermore, it is well established that pioneer TFs can employ various modes of nucleosome eviction, which may not occur directly at the target site of chromatin accessibility (7, 59). Finally, although there is a difference in the set of TFs that are responsible for regulating the chromatin accessibility of promoters and enhancers (60), we could show that for the general relationship of nucleosome positioning in the competition with TF binding, both groups show a similar behavior.

In contrast to these nucleosome positions determined by DNA sequence, constitutive heterochromatin, although commonly occupied by nucleosomes, does not demonstrate a comparable reliance on DNA sequence for precise nucleosome positioning. In analogy to the previous explanation of DNA evolution toward a targeted occlusion of TFBSs to prevent random gene activation, we propose a lower relevance of sequence for heterochromatin. The higher order packing of dense chromatin is a more unspecific process in which the accessibility of small individual positions on the genome is not expected to have such severe consequences as the revealing of TFBSs in cis-regulatory elements.

Apart from the evolution of nucleosome positioning advantage at highly competitive sites, we found other instances of locations that appear to have high sequence-intrinsic nucleosome support. In particular, exons show considerable nucleosome support relative to their surroundings. It is well described that the occupation of exon regions with nucleosomes is important in the pausing of RNAPII to ensure the correct transcription of a gene, as well as the marking and spatial relocation of exons for the splicing machinery (11, 61). Certain studies propose that the sequence specificity for nucleosomes at exons is primarily encoded at the exon's outset, specifically in the form of a specific dinucleotide distribution at the intron/exon junction (62). However, the positioning of the nucleosome based on the sequence is found in this study directly downstream of the exon start. This finding is in line with the theory that the nucleosome in the exon stalls the RNAPII just upstream of the IE junction (63). The sequence-encoded attraction of nucleosomes to exon locations might therefore be another example of reducing the energy needed to guide nucleosomes to consistently regulate RNAPII dynamics.

The principle of using the DNA to encode a favorable structure to guide nucleosomes to exon regions might again be an efficient means of increasing the intrinsic nucleosome support. In contrast to strengthening the nucleosome in competition with TFs, the nucleosome at exons might simply be guided to its place of effect without the need for further displacement by other processes. The difference in the non–sequence-dependent nucleosome positioning at the TSS

could be grounded in the reduced energy needed to stabilize nucleosomes permanently at exons compared with the potential for dynamic shifting of nucleosomes around TSS upon gene activation. This provides another striking example of putative co-evolution that is directed to attract nucleosomes on the sequence, which underlies a drive to reliably encode proteins at the same time.

The use of sequence to influence nucleosome positioning is evident not only in its ability to positively guide nucleosomes to specific locations. We showed the opposite effect for poly(A) sites, marking the end of transcription. The substantial contrast in sequence-intrinsic nucleosome support between poly(A) sites and the surrounding DNA clearly indicates the strategic prevention of nucleosome binding at these specific locations. The reason for this feature could be the attempt to keep chromatin at the TTS constantly accessible to easily allow the binding of proteins that form complexes together with the RNAPII to terminate transcription, cleave the mRNA, and do the polyadenylation in the consecutive step. The nucleosome rejection fits well with the general observation of low nucleosome occupancy over poly(A) sites. However, there are hints about the role of increased nucleosome occupancy around one end of the sites, regulating alternative polyadenylation and depending on the status of active transcription (13). This could be the reason for a difference in nucleosome support toward both directions surrounding poly(A) sites in terminal exons. Here, we observed slightly lower nucleosome support downstream of poly(A) sites for terminal exons but not as a general trend over all groups. Also, we could show an increased nucleosome support upstream of the TSS for the group of sites downstream of the TSS, in which the increase is easily explainable by the overlap with the generally sequence-defined promoter region. It cannot definitively be determined whether the otherwise relatively constant nucleosome occupancy up- and downstream arises from technical aspects in the normalization of ENCODE MNase-seq data or whether it is attributed to variations in the splicing patterns of different clusters of poly(A) sites, particularly when situated in regions other than terminal exons, or whether all of these sites possess transcription termination functionality at all.

Opposed to the steering of functional nucleosomes by nucleotide sequence, we identified processes that exhibit strong nucleosome positioning that does not manifest on the DNA sequence level. Notably, downstream of the TSS, a well-established pattern of regularly spaced nucleosome array formation is observed. At these locations, the nucleosome support constantly decreases toward intronic locations without mirroring the periodicity of the nucleosome occupancy. This process is known to be mediated by active chromatin remodeling complexes, which are organizing nucleosomes into an evenly spaced array using energy from ATP (10). This process serves as an example of an important transcriptional mechanism, which is not directly supported by local DNA structure. Regarding the nucleosome arrays adjacent to the TSS, it can be speculated that the +1 and −1 nucleosomes are influenced by sequence-specific influences. This assertion is supported by the relatively high NF score, aligning with studies showing the special significance of these particular nucleosomes (8, 64). These studies suggest a discernible dinucleotide enrichment in proximity to these nucleosome locations, implying a mechanism of "statistical positioning" for the co-occurring nucleosome arrays. However, the proximity of these nucleosomes to the main areas of TF–nucleosome competition in the promoter could easily lead to misinterpretation of their reliance on DNA sequence alone.

In conclusion, it is worth highlighting that these processes of sequence-intrinsic and sequence-independent nucleosome positioning may not necessarily be mutually exclusive. This phenomenon is particularly pronounced at insulator sites, where the transcription factor CTCF plays a crucial role in delineating TAD boundaries. Here, it could be observed that on top of the actual binding location there is very strong sequence-defined support for the nucleosome, suggesting the previously described mechanism of competition and co-evolution of TF–nucleosome interaction. Yet, as a result in the surroundings of CTCF-bound locations, there is a clear well-positioned nucleosome array, which is the result of active chromatin remodeling without the help of sequence support. Although the binding of CTCF underlies the aforementioned competition with positioned nucleosomes, the spacing and symmetry of the surrounding array are rather linked to active chromatin remodelers, instead (5).

This is just one example of the very close, local interplay of sequence support and energy-dependent remodeling of chromatin to decide the final in vivo positions of nucleosomes in transcriptional regulation. It is impossible to favor any of these principles in their importance for the collaborative regulation of transcription, from the formation of TAD boundaries, over the regulation of gene activity through the accessibility of chromatin for TFs or support mechanisms of the tasks of RNAPII, to the clean termination of transcription, thus marking the last step in which the DNA is involved directly.

# Materials and Methods

### Datasets

The experimental chromatin accessibility datasets were obtained from ENCODE for the human genome in hg19 (41). They comprise MNase-seq data as a density graph of signal enrichment, described in reference 65; DNase-seq data as read depth normalized signal, described in reference 66; and peak-called histone modification ChIP-seq data as bed narrowPeak locations for K562 and GM12878 and raw signal for H3K4me3 and H3K27ac as control normalized tag density, described in reference 67. Besides, we used CTCF ChIP-seq peaks from ENCODE for our insulator analysis. Furthermore, we used the comprehensive list of ChIP-seq narrowPeak locations for all TFs that are available on ENCODE for hg19. When referring to the central peak position related to the ENCODE data, we reference the point-source peak defined in the narrowPeak format standard. Because of their extensive length, we show the ENCODE IDs for all the aforementioned datasets in Tables S2, S3, and S4.

In addition, we used nucleosome occupancy data to evaluate the influence of the GATA3 TF in MDA-MB-23 cells with and without GATA3 expression. The according MNase-seq data were taken from Takaku et al (2016) (49). We calculated the mean of all three replicates for the condition of GATA3 expression and for cells without GATA3 expression.

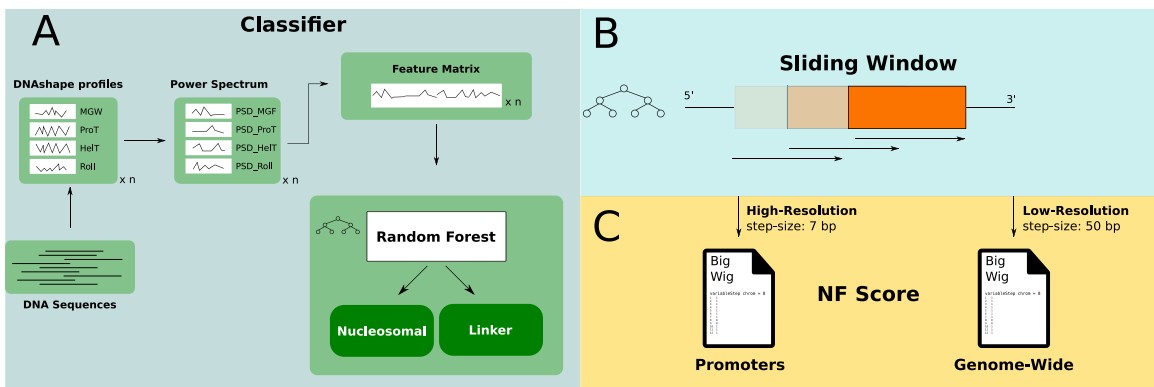

**Figure 6. Processing and classification of DNA sequences—deriving the NF score.**
**(A)** Random forest model is used to annotate DNA sequences as nucleosomal or linker. To build the feature matrix, DNAshape features are obtained for the input sequences and further transformed into a frequency power spectrum. These PSD profiles are then concatenated into the final input feature matrix to use in the random forest model for training and classification. **(B, C)** To create a pre-calculated mapping for the human genome, we applied the classifier genome-wide in a sliding-window approach with a step size of 50 bp and a detailed version for promoters with a step size of 7 bp. Both resources are provided as a downloadable bigWig file (see the Data Availability section).

Furthermore, we evaluated the evolutionary conservation at a base pair resolution with the PhyloP score (68), which was obtained using the UCSC Table Browser (45).

We compared all these data along a variety of different genome locations. The broad overview over different biological contexts is defined by chromHMM for K562 and GM12878 and defines 15 meaningful states originally derived from histone modification ChIP-seq data (44).

The gene mapping we use is based on RefSeq in hg19 (69) and was also obtained with the UCSC Table Browser. We exclude genes from chromosome Y, because not all experimental chromatin accessibility data are available for it. In particular, we use the remaining 30,142 unique TSS locations to define genes and promoters. Our promoter definition is defined as within 1-kb upstream of each TSS defined by RefSeq. In case of multiple alternative TSS locations, all of these were considered as individual gene starts. Besides using RefSeq for TSS, we also extracted all exon locations from the same mapping.

Moreover, we analyzed the elements at TTSs in the form of poly(A) sites. The poly(A) site mapping with a separation into clusters of biological origin is taken from PolyASite 2.0 (52). To obtain single locations for the center of each site, we used the representative cluster location from the unique cluster ID column. The original assembly of the data is hg38. We used liftOver to convert the coordinates to hg19 via the rtracklayer R package (70).

## Software

Apart from the individual software that is referenced in its due place, the analyses were performed using R (v. 4.3) (71), and we used the packages rtracklayer (v. 1.60.0) (70), ggplot2 (v. 3.4.2) (72), GenomicRanges (v. 1.52.0) (73), ROCR (v.1.0.11) (74), and random-Forest (v. 4.7.1.1) (75).

## Classifier

The classifier is based on the human Guo et al (2014) dataset (33), which is in turn derived from an in vivo mapping of nucleosome positions in CD4[+] T cells (39). It contains 4,573 non-redundant FASTA sequences, which are categorized as either nucleosomal (n = 2,273) or linker sequences (n = 2,300). A mapping of these sequences to the hg19 human reference genome is in addition provided as BED files (see the Data Availability section). The labeling strategy of the original authors is based on a ranking of sequences by a summation of the underlying nucleosome occupancy counts. Therefore, nucleosomal sequences are the most consistently occupied genome regions, whereas linker sequences are understood as stretches of DNA that do not show a nucleosome occupancy signal. Because nucleosomes usually bind 147 bp of DNA, the same length is used for the linker sequences, although depending on the definition, these can show different lengths in the genome. Fig 6 shows the workflow for setting up the classifier (Fig 6A), transforming the input data and its genome-wide application with a sliding-window approach (Fig 6B). The FASTA files are transformed with the DNAshapeR tool (v. 1.28.0) (43), resulting in a set of numerical vectors for Roll, HelT, ProT, and MGW with one value per bp each. The mean profiles of each DNAshape feature are compared between nucleosomal and linker sequences in Fig S7 (left). In addition, a power spectrum (PSD) is applied to the DNAshape values in order to strengthen positional independence of the periodic nucleotide patterns, emphasizing the assumed underlying periodic sequence pattern and making it suitable for the use in a random forest classifier. That step uses the spectrum function in R with a taper of 0.1 and a span of 10. In the next step, the resulting spectral densities are concatenated into a feature matrix, which is the input for a random forest classifier. The input features for the classifier consist of the PSD values per frequency for each DNAshape feature. Inside the model, features are denoted by a combination of the DNAshape feature and frequency. For instance, the input feature HelT_5 represents the power of the underlying HelT sequence at a frequency of 5 per 1/147 bp. A high power for low-frequency numbers indicates a significant contribution from long-range signals. Fig S7 visualizes the PSD profiles for nucleosomal and linker DNA. The random forest classifier was trained with the randomForest package in R with default parameters. The result of the

**Table 2. Performance measures for the random forest classifier, evaluated with 10-fold cross-validation.**

|  | Performance |
|---|---|
| Accuracy | 82.01% (±1.55) |
| Sensitivity | 85.36% (±2.29) |
| Specificity | 78.93% (±3.26) |
| F1 | 85.10% (±1.88) |
| AUC | 0.89 (±0.01) |
| MCC | 0.64 (±0.03) |

The numbers in brackets denote the SD of these measures within the 10-fold.

classification is the categorization of a sequence being either *nucleosomal* or *linker* and serves as the basis for the two types of NF scores that will be described further in the NF score section. The particular final number of input features at the end of the transformation is 288. This results from the basic sequence lengths (147 bp), the number of DNAshape parameters used for training (4), and a subtraction of some missing DNAshape values at the edges of the sequence where the centered k-mer mask is incomplete (12). These 576 features are halved because of the redundant nature of both sides of a full PSD spectrum to the final number of 288 input features.

The performance of the random forest classifier based on 10-fold cross-validation is shown in Table 2. The accuracy is measured with 82.01% (±1.55) and the AUC with 0.89 (±0.01). Furthermore, we compiled a range of different random forest models based on all of the previously mentioned data pre-processing steps. Table S5 lists performance measures for classifiers trained on raw sequences, pure DNAshape values, and the final PSD frequency transformation. It demonstrates that with each further transformation, there is a substantial gain in performance, which underlines the suitability of the frequency modeling to cover the underlying periodicity proposed by Segal et al (2006) (20).

The most recent classifiers use CNNs and reach accuracies of up to 0.889 (CORENup (76)) and an AUC of up to 0.94 (LeNup (35)) on the given dataset. However, although CNNs are undoubtedly valuable for such prediction tasks in principle, we opted not to use them on this specific dataset because of their propensity for overfitting on smaller datasets. Furthermore, the random forest methodology enables us intrinsically to derive feature importance scores from the model and we have a direct connection between the NF score resource and what determines the score on the basis of periodic DNA structure changes in the form of PSD. The feature importance was obtained using the importance function from the randomForest package in R and is provided as a mean decrease in Gini impurity. The ranking of each input feature by importance can be viewed in Table 1.

The present in vivo training dataset is a common standard in the prediction of nucleosome positioning. We want to point out that the use of in vivo data has the potential to introduce bias in predicting nucleosome positioning events, independent of the chosen classification method. Although previous studies have demonstrated the promise of incorporating in vitro nucleosome maps (55, 77), we find it noteworthy that despite training on in vivo data, we can show that the classifier is identifying the underlying sequence importance rather than the in vivo nucleosome occupancy that is resulting from the dynamic displacement of sequence-positioned nucleosomes in living cells. This is especially evident in the first section of the results about the counteracting relationship of high sequence-intrinsic nucleosome support in regulatory important regions against the low actual nucleosome occupancy in vivo at these positions of nucleosome–TF competition.

## Score calculation

### Nucleosome formation (NF) score

The classifier described above gives a binary prediction for a DNA segment of 147 bp in length. To allow for easier handling of the estimates of nucleosome support based on the nucleotide sequence, we pre-calculated two sets of scores for the human genome in hg19 assembly with different local resolutions as seen in Fig 6C. (1) The low-resolution score is a simple application of the classifier in a sliding window to all regions of the human genome with a fragment input length of 147 bp and a step size of 50 bp. The resulting NF score is an estimate of either 1 (nucleosomal) or 0 (linker) for the central 50 bp in any given 147-bp bin. (2) The high-resolution NF score provides a continuous score between 0 and 1 and is based on a much more local sliding window with a segment size of 147 bp and a step size of 7 bp. All base pairs are thus part of multiple (147/7 = 21) predictions, and the score represents the relative frequency of nucleosomal prediction results that this particular bp has been part of. To give an example, if an individual bp has been part of 21 sliding windows, of which 14 were classified as nucleosomal and 7 as linker, then the score at that base pair is 14/21 = 0.67. Because of the introduction of artifacts, otherwise the step size needs to be a divisor of the window size of 147 bp. Thus, we chose a resolution of 7 bp as it is the most local one next to a resolution of 1 bp.

Because of resource efficiency, the high-resolution score was only pre-calculated for all human promoter regions, because we want to give a good local estimate of the importance of the sequence for nucleosome positioning in specific individual promoter regions. In contrast to the high-resolution score, the step size of 50 for the low-resolution score is purely pragmatic because there is no further integration of the classification results. If not specified otherwise, we always use the low-resolution score, because we normally show the mean of nucleosome support profiles over a large number of fragments and there is virtually no difference between averaging the low-resolution 0/1 predictions and doing the same with the 0–1 high-resolution score. To illustrate the sufficiency of the 50-bp low-resolution score when examining larger sets of data in an overview fashion, we sampled 10,000 of the promoters used in this study randomly and calculated mean profiles with both the high- and the low-resolution scores and compared their similarity in Fig S11. The difference between both profiles varies maximally between 0.01 and 0.015. Both resources are easily accessible as bigWig files and listed in the Data Availability section. In addition, the code to classify nucleosomal versus linker DNA can be found there. The NF score described in this section is what is in this work referred to as sequence-intrinsic nucleosome support.

*DHS score*

In this study, we analyze the influence of sequence on nucleosome positioning, However, the resulting accessibility in particular cell lines can be very cell type–specific. Therefore, we created a score to generalize the chromatin accessibility over multiple cell lines. The DHS (DNase hypersensitive sites) score is a measure to estimate the generalized accessibility of any given genome location. For that purpose, we obtained peak-called DNase-seq data for all 403 primary cell lines from ENCODE in humans. These were integrated as a simple count of how many primary cell lines show a DNase-seq peak at any given base pair.

To give an example, a score of 403 at a specific bp means that all of the available cell lines show a DNase-seq peak overlapping that position. A score of 5 means that only five specific cell lines show accessible chromatin in the form of a DNase-seq peak at the specified bp. In the study, the score is mostly used in the context of indicating TF binding.

## Data Availability

The code to execute predictions can be found at https://gitlab.gwdg.de/MedBioinf/generegulation/nfclassifier. The NF scores in bigWig file format can be downloaded at https://bioinformatics.umg.eu/resources/nfscore. The Guo et al training dataset was further mapped to the human reference genome hg19 in the form of BED files using the Rsubread R package (78). These mappings are appended as additional Supplemental Data 1 and 2, separated by nucleosomal and linker sequences.

## Supplementary Information

## Acknowledgements

We thank Argyris Papantonis and Alexander Ecker for fruitful discussions. M Sahrhage is a member of the International Max Planck Research School for Genome Science (IMPRS-GS), part of the Göttingen Graduate Center for Neurosciences, Biophysics, and Molecular Biosciences (GGNB). NB Paul is a member of the Molecular Medicine PhD program, part of the Göttingen Graduate Center for Neurosciences, Biophysics, and Molecular Biosciences (GGNB). T Beißbarth and M Haubrock are members of the Göttingen Campus Institute Data Science (CIDAS). Also, we thank the ENCODE Consortium and the ENCODE production laboratories for generating the particular datasets. The research was funded by Deutsche Forschungsgemeinschaft (DFG) (KFO5002 to M Sahrhage and SFB1002 to NB Paul). We acknowledge support by the Open Access Publication Funds/transformative agreements of the Göttingen University.

### Author Contributions

M Sahrhage: conceptualization, resources, data curation, software, formal analysis, validation, investigation, visualization, methodology, project administration, and writing—original draft, review, and editing.

NB Paul: conceptualization, software, and writing—review and editing.

T Beißbarth: resources, supervision, funding acquisition, methodology, and project administration.

M Haubrock: conceptualization, data curation, software, supervision, methodology, project administration, and writing—review and editing.

### Conflict of Interest Statement

The authors declare that they have no conflict of interest.

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
