## [Reviewer comments · Life Science Alliance]

Life Science Alliance

The importance of DNA sequence for nucleosome positioning in transcriptional regulation

Malte Sahrhage, Niels Paul, Tim Beissbarth, and Martin Haubrock

DOI: <https://doi.org/10.26508/lsa.202302380>

Corresponding author(s): Malte Sahrhage, University Medical Center Goettingen and Martin Haubrock, University Medical Center Goettingen

Review Timeline:

Submission Date:	2023-09-18
Editorial Decision:	2023-10-25
Revision Received:	2024-02-09
Editorial Decision:	2024-03-05
Revision Received:	2024-05-08
Editorial Decision:	2024-05-09
Revision Received:	2024-05-15
Accepted:	2024-05-16

Transaction Report:

October 25, 2023

Re: Life Science Alliance manuscript #LSA-2023-02380-T

Malte Sahrhage
University Medical Center Goettingen
Department of Medical Bioinformatics
Germany

Dear Dr. Sahrhage,

Thank you for submitting your manuscript entitled "The importance of DNA sequence for nucleosome positioning in transcriptional regulation" to Life Science Alliance. The manuscript was assessed by expert reviewers, whose comments are appended to this letter. We invite you to submit a revised manuscript addressing the Reviewer comments.

Thank you for this interesting contribution to Life Science Alliance. We are looking forward to receiving your revised manuscript.

Sincerely,

B. MANUSCRIPT ORGANIZATION AND FORMATTING:

Reviewer #1 (Comments to the Authors (Required)):

The article addresses an important problem of the role of DNA sequence in nucleosome positioning. A slight weakness is that it uses a bit outdated experimental dataset for the training, while more high-resolution datasets are available now, but this should not be a problem for the computational analysis and conclusions. The Results include interesting observations about sequence-encoded nucleosome binding at TF binding sites and well as nucleosome positioning story for exons and poly-A tail regions. I have reviewed a previous version of this manuscript submitted to NAR, and I can see that the authors have addressed most of my comments. Therefore, I have only minor comments here:

- 1) Figure 1C - please add captions to the X and Y axes
- 2) Figure 3B - I am not sure why NF correlates with the DHS and DNase-seq signal, while it's expected to correlate with the MNase-seq signal. It would be good if the authors include a comment on this in the updated manuscript.
- 3) Figure 6 - Could the authors please comment, why NF anti-correlates with MNase at TSS, but correlates at exons? Also, what happens at the first exon (which is essentially at the TSS)?
- 4) The description of the code could be a bit expanded.

Reviewer #2 (Comments to the Authors (Required)):

The manuscript by Sahrhage, Paul, et al., aims to provide insight into the sequence basis for nucleosomal positioning in gene regulation. The authors use a previously published set of nucleosomal sequences and "linker" sequences to train a random forest classifier to generate a nucleosome formation (NF) score that indicates the propensity of a given sequence to be nucleosomal.

Through genome wide predication of the NF score and comparison with a large number of ENCODE datasets the authors derive conclusions on the nucleosome sequence preferences at enhancers and promoters in two cell lines. The principal conclusion of the work, shown over a number of examples, is that the author's NF score is highest at regions enhancers, promoters, and overall regions bound by transcription factors (TF). For example, a peak in the NF score coincides with the nucleosomal depleted region at many TF bound loci, the author's argument being that these sequences intrinsically position nucleosomes to compete with TFs. Additionally, that nucleosomal sequence preference is predicted at exon start and polyA sites.

These are interesting observations, if they are substantiated.

However, I have significant concerns over the methodology, particularly what are the origins of the sequences used to train the random classifier. Moreover, the manuscript contains overreaching discussion in places, that should be focused more on the content of the paper.

Major concerns on construction of the classifier and methodology:

The author's should supply additional confidence that their classification is identifying nucleosomal preference as opposed to some other sequence feature.

Methods state that the classifier is trained on nucleosomal and linker sequences. Reviewing the Guo et al. 2014, it appears that the sequences derive from chromosome 20, with nucleosomal sequences derived by Schones et al., 2008 MNase (25bp). The "linker" sequences appear to be 147bp non-nucleosomal DNA defined by Guo et al., 2014, however, the precise definition of these is not obvious. Moreover, the phrase linker sequence is commonly used to refer to short (~40bp) sequences between nucleosomes in an array. This terminology should be corrected throughout this paper.

As is not obvious from Guo et al., 2014 how the non-nucleosome sequences are defined, and since all sequences derived from Guo et al., 2014 are critical to the all author's observations, the author's should map these sequences to the genome and be

explicit as to where they derive, scoring them against chromatin features such as the used chromhmm chromatin states.

Ideally, the authors should use their approach with a more recent MNase than 2008 that are substantially deeper, use paired end sequencing so that precise nucleosomal fragments can be identified, and then train on a subset of chromosomes with others held out. However, I would be willing to consider a revision of the paper with further confidence supplied in the current dataset.

In addition to this, the authors should supply further insight into what the random forest classifier has learnt through feature importance scores, how does their approach compare to other approaches on the same data? For example, is it correct that the Guo et al., 2014 has a larger AUC of 0.925 as compared to 0.883 achieved here. The authors should give some discussion as to the definition and meaning of the DNA shape features, and give an indication of an example sequence and DNA properties preferred by nucleosomes and otherwise. Any further context such as comparisons of AUC to baseline models such as nucleotide content, di-nucleotide content or short k-mer content would also be very useful.

Other issues:

- Given the overall concerns in the literature on the meaning of spatial separation in dimensionality reduction approaches such as UMAP and t-SNE, I am not sure Fig 1C and accompanying discussion adds much value, and would prefer to see more details on the random forest model in its place.
- The conclusion that promoters are preferred in K562 and enhancers in GM12878, seems overreaching with no replicates.
- The text and figure should describing the use of the GATA3 positive and negative MNase, should be made more clear to indicate which cells are measured and that the cells are +GATA3 and -GATA3. Some of this text is present in the methods, but it should be made more explicit in the main text.
- CTCF peaks and its array are not strand agnostic. CTCF binds to one or other strand, and the nucleosome array has been shown to be slightly asymmetric around the strand of the CTCF motif. Given that this asymmetry is minor, it sentence on the strand of peaks can be dropped.
- The methods on how public datasets were used is insufficient. For the MNase datasets used, it should be made clear how the data reflects nucleosomal positioning. ie single-end, paired-end, how is it smoothed, units etc. It isn't obvious how to interpret the given ref (44), if this is a URL please supply that.
- The word "chapters" at the end of the methods appears to refer to something beyond the paper.
- The exon and polyA site results seem striking - can the authors confirm that they are excluding first exons (ie TSS) from their exon starts results. Given the point about about short exons, the authors should either stratify their exon mid summary by exon size, or refine which exons are included to ensure the signal is distinct from exon starts.

Response to Reviewers

Reviewer #1

The article addresses an important problem of the role of DNA sequence in nucleosome positioning. A slight weakness is that it uses a bit outdated experimental dataset for the training, while more high-resolution datasets are available now, but this should not be a problem for the computational analysis and conclusions. The Results include interesting observations about sequence-encoded nucleosome binding at TF binding sites and well as nucleosome positioning story for exons and poly-A tail regions. I have reviewed a previous version of this manuscript submitted to NAR, and I can see that the authors have addressed most of my comments. Therefore, I have only minor comments here:

We appreciate the reviewer for re-evaluating the manuscript and recognizing the modifications made in the current version. Furthermore, we are thankful for the support of our story and the new comments in this review to help us clarify some of the main findings in the main text

Minor comments

Figure 1C - please add captions to the X and Y axes

In response to another reviewer's suggestion, we removed the t-SNE plot from Figure 1C as it was deemed to contribute minimal information. The primary objective of the subfigure was to demonstrate the separability of the datapoints in principle. However, this is evident from the classifier performance as well as a new image in the supplementaries (Supplementary Figure S7), in which the training set is visualized both by DNashape and PSD frequency values.

Figure 3B - I am not sure why NF correlates with the DHS and DNase-seq signal, while it's expected to correlate with the MNase-seq signal. It would be good if the authors include a comment on this in the updated manuscript.

We thank the reviewer for bringing up this point, since it is an essential message to further clarify in the main manuscript. The NF score correlates positively with the MNase-seq signal to initially position important regulatory nucleosomes for inactivation of gene promoters. When these nucleosomes are displaced upon gene activation, the nucleosome is still supported at the initial position, but since the nucleosome is then shifted from the position and TFs bind the same location, the region is then occupied by TFs and the NF score correlates with the DNase-seq signal instead.

We added a sentence at the end of the description summary to make this clearer in the main text.

[...] Fig 2A and 2B together lead us to say that locations in promoters where TFs compete with nucleosomes directly are the places with the highest nucleosome support. Whether these are occupied by nucleosomes is dependent on whether the gene is actually expressed. Thus, we are not of the opinion that all nucleosomes in the promoter are sequence-positioned, as suggested in the mean of all promoters in Fig 2A. In these overall promoter regions, the NF score stays constantly relatively high but in the individual promoters in Fig 2B, it depends on the co-localization with competing TFs that mediate chromatin accessibility at local, defined places. **The NF score exhibits a positive correlation with nucleosome occupancy in inactivated genes at regulatory key nucleosomes, where the nucleosome is deliberately positioned by sequence. In contrast, it correlates negatively once the corresponding nucleosome is displaced by targeted TF action at the same location.**

Furthermore, we hope to clarify this point further in the explanation of co-evolution of sequences to support both nucleosomes and the counteracting TFs at the same position but on different frequencies that we derive from feature importance scores and describe in a new part of the Discussion that relates to a new section of the end of the results named "Low-Frequency differences in HeIT are the most discriminative sequence feature for nucleosome positioning".

[...] Therefore, we interpret the crucial function of DNA in these "regulatory" nucleosomes as a mechanism to regulate the competition between nucleosomes and TFs for spatial availability. This competition affects the energy required to activate biological processes, allowing for dynamic control and modulation. **This co-localized competition raises the question of how sequence-encoded nucleosome preferences and well-defined TFBSs can evolve at the same positions. The answer lies in the analysis of feature importance derived from the random forest model. This study demonstrates that the primary predictors for distinguishing between nucleosomal and linker DNA stem from low-frequency, i.e., long-range, differences in HeIT. Simultaneously, it is well-documented that TFs recognize very local, high-resolution, i.e., high-frequency, DNA structure signals [52]. This leads to the assumption that the nucleosome attraction and the counteracting TFs are encoded at the same positions but on different frequencies of DNA structure changes. [...]**

Figure 6 - Could the authors please comment, why NF anti-correlates with MNase at TSS, but correlates at exons? Also, what happens at the first exon (which is essentially at the TSS)?

Due to rearranging the position of the Methods and Materials section, the Figure discussed here is now Figure 5 instead of 6.

The correlation between NF and MNase-seq data: Upstream of the TSS, there is an anti-correlation between the NF score and MNase-seq data, primarily for the same reasons outlined in the preceding point. These positions represent instances where nucleosomes are drawn to the sequence (high NF) but are rendered accessible through the action of transcription factors. Downstream of the TSS, the NF score decreases while nucleosome occupancy is relatively high. This means that the nucleosomes there are positioned but not by sequence preferences. This is underlined above all by the observation that the

nucleosomes are positioned in a well-defined array, whose periodic oscillation is not mirrored on sequence level.

The correlation at exons is evident in the figure since our NF score mirrors the occupancy perfectly. With our separate view of exons aligned by start (IE junction) or mid positions, we relate the preferences to an evolutionary favored way of reducing the energy for actively placing exons on top of exons to slow down RNAPII transcription speed for improving transcription accuracy. We added a similar part to enhance the corresponding paragraph in the Discussion:

[...] Apart from the evolution of nucleosome positioning advantage at highly competitive sites, we found other instances of locations that appear to have a high sequence-intrinsic nucleosome support. In particular, exons show considerable nucleosome support relative to their surroundings. It is well described that the occupation of exon regions with nucleosomes is important in the pausing of RNAPII to ensure the correct transcription of a gene as well as the marking and spatial relocation of exons for the splicing machinery [11], [59]. Certain studies propose that the sequence specificity for nucleosomes at exons is primarily encoded at the exon's outset, specifically in the form of a specific dinucleotide distribution at the Intron/Exon junction [60]. However, the positioning of the nucleosome based on the sequence is found in this study directly downstream of the exon start. This finding is in line with the theory that the nucleosome in the exon stalls the RNAPII just upstream of the IE junction [61] **The sequence-encoded attraction of nucleosomes to exon locations might therefore be another example of reducing the energy needed to guide nucleosomes to consistently regulate RNAPII dynamics.** [...]

First exons: The question of first exons is indeed interesting in this context. In the main manuscript, we combine all exons in the image. To distinguish between first and all other exons, we added Supplementary Figure S5, in which we separate these two classes. It is evident that the image for all exons except the first exhibits the same shape as the one in the main manuscript due to the relative minority of first exons. However, the individual profiles of first exons do not exhibit the same nucleosome occupancy but are instead marked by an absence of nucleosomes over the midpoints and a clear occupation signal at the supposed +1 position. This result is referred to in the manuscript at the last sentence of the “Sequence-intrinsic nucleosome occupation potential of exon regions” subsection. This is a screenshot of the new supplementary image:

Supplementary Figure S5. First Exons. MNase-seq and NF scores for all human exons except the first (top) and for only the first exons, essentially the TSS (bottom). It is evident that the exons excluding all first exons are characterized by nucleosome occupancy over their midpoints while first exons show only a clearly defined downstream nucleosome which can be interpreted as the +1 nucleosome.

The figure is referenced also in the main manuscript with this sentence in the exon section:

[...] The images shown here encompass all human exons. Since the first exon is essentially the TSS, Supplementary Fig S5 shows the same profiles, once for all exons except the first of each gene and once only for the first exons. Due to their comparatively small amount, the principal image does not change when filtering out the first exons. The isolated first exons, however, do not exhibit the same occupation of exons but rather show the same nucleosome positioning as shown in the TSS analysis in Fig 5A with the clear +1 nucleosome.

The description of the code could be a bit expanded.

We acknowledge the reviewer's comment and interpret the suggestion as pertaining to the manuscript itself, rather than the GitLab repository.

We aimed for addressing the comment on different levels in the "Classifier" paragraph of the Material and Methods section. First of all, we tried to clarify the data flow within our pre-processing scripts. In particular, we added details about the setup/labeling strategy of the training data, the dimensionality of DNashape values and PSD profiles and a clarification of what the PSD values represent.

The classifier is based on the human Guo et al. 2014 dataset [33] which is in turn derived from an *in vivo* mapping of nucleosome positions in CD4⁺ T-cells [39]. It contains 4573 non-redundant FASTA sequences which are categorized as either nucleosomal (n=2273) or linker sequences (n=2300). The labeling strategy of the original authors is based on a ranking of sequences by a summation of the underlying nucleosome occupancy counts. Therefore, nucleosomal sequences are the most consistently occupied genome regions, while linker sequences are understood as stretches of DNA that do not show a nucleosome occupancy signal. Since nucleosomes usually bind 147 bp of DNA, the same length is used for the linker sequences, although depending on the definition, these can show different lengths in the genome. Fig 6 shows the workflow for setting up the classifier, transforming the input data and its genome-wide application with a sliding window approach. The FASTA files are transformed with the DNashapeR tool (v. 1.28.0) [43], resulting in a set of numerical vectors for Roll, HelT, ProT and MGW with one value per bp each. The mean profiles of each DNashape feature are compared between nucleosomal and linker sequences in Supplementary Fig S7 (left). Additionally, a power spectrum (Power Spectral Density, PSD) is applied to the DNashape values in order to strengthen positional independence of the periodic nucleotide patterns, emphasizing the assumed underlying periodic sequence pattern and making it suitable for the use in a random forest classifier. That step uses the spectrum function in R with a taper of 0.1 and a span of 10. In the next step, the resulting spectral densities are concatenated into a feature matrix which is the input for a random forest classifier. The input features for the classifier consist of the PSD values per frequency for each DNashape feature. Inside the model, features are denoted by a combination of the DNashape feature and frequency. For instance, the input feature HelT_5 represents the power of the underlying HelT sequence at a frequency of 5 per 1/147 bp. A high power for low frequency numbers indicates a significant contribution from long-range signals. Supplementary Fig S7 visualizes the PSD profiles for nucleosomal and linker DNA. The random forest classifier was trained with the randomForest package in R with default parameters. The result of the classification is the categorization of a sequence being either *nucleosomal* or *linker* and serves as the basis for the two types of NF scores that will be described further in the NF score section.

To accompany these notions visually, we added the aforementioned figure in the Supplementary Material (S7) with the mean profiles of both DNashape raw values and PSD transformed profiles for nucleosomal and linker sequences.

Supplementary Figure S7. Visualization of the training data. Left: Average profiles of DNashape values over all training samples from the Guo et al. set of linker vs nucleosomal sequences. Right: Average profiles of the power spectral density (PSD) for each of the DNashape features. The datasets appear separable in principle. The PSD profiles are more different between both groups on low frequencies.

Also, we added Supplementary Table S5, in which we show the progression of increasing classifier performance for training the model with the raw DNA sequences, the DNashape profiles and the PSD profiles. It can be seen that our transformation steps increase the classifier performance with each transformation step.

Supplementary Table S5. Performance for different stages of training data pre-processing, evaluated with 10-fold cross-validation. The numbers in parentheses denote the standard deviation of these measures within the 10 folds. Each transformation step exhibits a gain in performance

	Raw Sequence	DNashape	PSD Spectrum
Accuracy	71.24 % (± 1.88)	78.38 % (± 1.76)	82.01 % (± 1.55)
Sensitivity	72.12 % (± 4.16)	78.38 % (± 3.35)	85.36 % (± 2.29)
Specificity	70.58 % (± 3.41)	78.42 % (± 1.28)	78.93 % (± 3.26)
F1	69.70 % (± 2.40)	78.45 % (± 1.91)	85.10 % (± 1.88)
AUC	0.79 (± 0.03)	0.85 (± 0.01)	0.89 (± 0.01)
MCC	0.43 (± 0.05)	0.57 (± 0.03)	0.64 (± 0.03)

Last but not least we added a paragraph to describe how we obtain feature importance measures to explain what input features are most important for the classification of nucleosomal DNA.

[...] Furthermore, the random forest methodology enables us intrinsically to derive feature importance scores from the model and we have a direct connection between the NF score resource and what determines the score on the base of periodic DNA structure changes in the form of PSD. The feature importance was obtained using the importance function from the randomForest package in R and is provided as mean decrease in Gini impurity. The ranking of each input feature by importance can be viewed in Table 1.

The results of the feature importance analysis make up the new chapter “Low-Frequency differences in HeT are the most discriminative sequence feature for nucleosome positioning” at the end of the results part.

Reviewer #2

The manuscript by Sahrhage, Paul, et al., aims to provide insight into the sequence basis for nucleosomal positioning in gene regulation. The authors use a previously published set of nucleosomal sequences and “linker” sequences to train a random forest classifier to generate a nucleosome formation (NF) score that indicates the propensity of a given sequence to be nucleosomal.

Through genome wide predication of the NF score and comparison with a large number of ENCODE datasets the authors derive conclusions on the nucleosome sequence preferences at enhancers and promoters in two cell lines. The principal conclusion of the work, shown over a number of examples, is that the author’s NF score is highest at regions enhancers, promoters, and overall regions bound by transcription factors (TF). For example, a peak in the NF score coincides with the nucleosomal depleted region at many TF bound loci, the author’s argument being that these sequences intrinsically position nucleosomes to compete with TFs. Additionally, that nucleosomal sequence preference is predicted at exon start and polyA sites.

These are interesting observations, if they are substantiated.

However, I have significant concerns over the methodology, particularly what are the origins of the sequences used to train the random classifier. Moreover, the manuscript contains overreaching discussion in places, that should be focused more on the content of the paper.

We are grateful to the reviewer for their time and interest in evaluating our paper. The feedback received indicates a thorough understanding of our main findings, which we appreciate. We believe that the additional work on input data and signal sequence interpretation will help address any remaining concerns.

Major comments

Major concerns on construction of the classifier and methodology:

The author’s should supply additional confidence that their classification is identifying nucleosomal preference as opposed to some other sequence feature.

We acknowledge the need for arguments to supply confidence in the classifier’s suitability to predict nucleosomal preferences. Our confidence stems from the fact that the only criterion for labeling the sequences in the original training set is their nucleosome occupancy in the form of MNase-seq signal without any further filtering steps. The ranking of the sequences by their nucleosome occupancy signal for labeling provides relatively clear well-positioned nucleosomes, particularly in comparison to the other main literature dataset in this context by Liu et al. 2014 (discussed in major comment #4). To make this clear for the example of regions that are targeted by TFs, naturally these positions have an inherent consistent DNA

pattern since they also host TF binding sites. However, that they are classified as nucleosomal rather than linker sequences implies that the corresponding locations are much more part of the foreground. Keeping in mind the ranking by nucleosome occupancy, this means that especially these regulatory nucleosomes are characterized by extraordinarily well-positioned nucleosomes. The confidence in the universality of the linker sequences can be seen in the Poly(A) example. As laid out in the next comment reply, linker sequences must be characterized by not showing nucleosome occupancy at all in the training set and it is interesting that we see such a clear drop in the NF score of these termination signals although it can be assumed that there is quite a wide range of different sequence locations in the training samples with no occupancy signal that do not originate from Poly(A) sites.

However, we agree that the labeling strategy of Guo et al. is not immediately accessible. Nevertheless, combining the information of Guo et al. and the source paper of Schones et al. 2018, it is possible to describe and understand the labeling strategy well enough to draw conclusions. We added more information about this labeling strategy in the Classifier section of the Material and Methods part. Since this is directly related to the next comment, we reference the related manuscript parts there.

Additionally to provide a specific notion of how the structure differs between both groups in the original training data, we added Supplementary Figure S7 to present mean profiles of linker and nucleosomal sequences in the structure and the frequency space.

Supplementary Figure S7. Visualization of the training data. Left: Average profiles of DNashape values over all training samples from the Guo et al. set of linker vs nucleosomal sequences. Right: Average profiles of the power spectral density (PSD) for each of the DNashape features. The datasets appear separable in principle. The PSD profiles are more different between both groups on low frequencies.

Methods state that the classifier is trained on nucleosomal and linker sequences. Reviewing the Guo et al. 2014, it appears that the sequences derive from chromosome 20, with nucleosomal sequences derived by Schones et al., 2008 MNase (25bp). The “linker” sequences appear to be 147bp non-nucleosomal DNA defined by Guo et al., 2014, however, the precise definition of these is not obvious. Moreover, the phrase

linker sequence is commonly used to refer to short (~40bp) sequences between nucleosomes in an array. This terminology should be corrected throughout this paper.

The critique of the nomenclature is justified. However, when using such sequences in machine learning, the same input dimensions need to be present for sequences from both groups. It is correct that Guo et al. derive their sequences in turn from Schones et al. 2018 from chr20 of CD4⁺ T-Cells. The 25 bp reads from the underlying experiment are integrated into an occupancy score by Schones et al., 2008. This score is a simple sliding window count, in which reads that fall within 160 bp around the individual location are summed up. This scoring enables Guo et al., 2014 to rank sequences by using the sum of these scores for 147 bp sequences. Presumably to get rid of neighboring sequences that are too similar (since they should have extremely similar counts), the authors use a software called CD-HIT to get rid of sequences that are more than 80 % similar. The top and bottom of the ranking are used to label sequences as nucleosomal or linker. Given this information, the linker sequences should be locations that are consistently free of nucleosomal occupancy and thus show a signal of 0 for 147 bp. Therefore, the definition of linker in this context and in our study is any part of interconnecting DNA that is not occupied by nucleosomes rather than short stretches of DNA within a clear nucleosome array and the length of the input DNA is 147 bp, although a wide range of different lengths could be possible.

To make these differences in nomenclature and the general labeling strategy more clear in the main text, we added a shorter version of the above description to the Classifier section of the Material and Methods:

The classifier is based on the human Guo et al. 2014 dataset [32] which is in turn derived from an *in vivo* mapping of nucleosome positions in CD4⁺ T-cells [38]. It contains 4573 non-redundant FASTA sequences which are categorized as either nucleosomal (n=2273) or linker sequences (n=2300). **The labeling strategy of the original authors is based on a ranking of sequences by a summation of the underlying nucleosome occupancy counts. Therefore, nucleosomal sequences are the most consistently occupied genome regions, while linker sequences are understood as stretches of DNA that do not show a nucleosome occupancy signal. Since nucleosomes usually bind 147 bp of DNA, the same length is used for the linker sequences, although depending on the definition, these can show different lengths in the genome. Fig 6 shows the workflow for setting up the classifier, transforming the input data and its genome-wide application with a sliding window approach. The FASTA files are transformed with the DNashapeR tool (v. 1.28.0) [42], resulting in a set of numerical vectors for Roll, HelT, ProT and MGW with one value per bp each. The mean profiles of each DNashape feature are compared between nucleosomal and linker sequences in Supplementary Fig S7 (left). [...]**

As is not obvious from Guo et al., 2014 how the non-nucleosome sequences are defined, and since all sequences derived from Guo et al., 2014 are critical to the all author's observations, the author's should map these sequences to the genome and be explicit as to where they derive, scoring them against chromatin features such as the used chromhmm chromatin states.

We thank the reviewer for this advice. However, the mapping of nucleosomes to the chromHMM mapping in the first results figure was only done to get a first idea of where

nucleosomes are on average more likely positioned by sequence. Considering the individual profiles investigating particular locations such as the promoters in Figure 2B (former Figure 3B), it is evident that not all nucleosomes are positioned by sequence even in these overall regulatory regions. We have to assume that the sequence preferences are determined by very local preferences of individual nucleosome occupancy peaks in the underlying training data. The nucleosome occupancy is the only labeling criterion. As such this is an unbiased approach which should equally cover linker sequences from all contexts within the limits of MNase-seq experiments which are admittedly not completely free of biases due to higher order chromatin packing influences. Overall, we did not observe results to suggest that the distribution of nucleosomal or linker sequences might be falsely over- or underrepresented in certain genomic locations but we definitely agree that more transparency in the labeling and the cutoff of the training sample nucleosome occupancy ranking would be helpful.

Ideally, the authors should use their approach with a more recent MNase than 2008 that are substantially deeper, use paired end sequencing so that precise nucleosomal fragments can be identified, and then train on a subset of chromosomes with others held out. However, I would be willing to consider a revision of the paper with further confidence supplied in the current dataset.

That is good advice and we indeed tried a range of newer datasets and labeling strategies by integrating data from multiple different chromatin accessibility experiments with promising results. However, we wanted to use an existing literature dataset in order to focus more on the application and interpretation of the measures rather than introducing a new method or dataset in this paper. In this context, most tools are compared on the current dataset of Guo et al, 2014 or on a related dataset of Liu et al. 2014 (chr1) (<https://doi.org/10.1093/bib/bbt062>), both of which we considered before attempting the analysis. They are derived from the same original experiment but differ in size and in labeling strategy. The Liu et al. dataset is less strict in its labeling of nucleosomal sequences but larger, making it more suitable for approaches using CNNs without overfitting. We trained a range of such machine learning architectures on that other training set. Despite its larger size, we decided to stick with the current dataset, since their strategy of ranking sequences by their occupancy proved well for reducing the number of false positive nucleosomal predictions. In contrast, the Liu dataset is labeled by defining nucleosomal labels as soon as 120 bp show any consecutive occupancy signal. Since these less strictly chosen nucleosomes occur everywhere in the genome but do not represent the sequence-defined, well-positioned nucleosomes that we try to identify, the resulting classifiers show a significant positive classification bias towards nucleosomal predictions.

In addition to this, the authors should supply further insight into what the random forest classifier has learnt through feature importance scores, how does their approach compare to other approaches on the same data? For example, is it correct that the Guo et al., 2014 has a larger AUC of 0.925 as compared to 0.883 achieved here. The authors should give some discussion as to the definition and meaning of the DNA shape features, and give an indication of an example sequence and DNA properties preferred by nucleosomes and otherwise. Any further context such as

comparisons of AUC to baseline models such as nucleotide content, di-nucleotide content or short k-mer content would also be very useful.

Feature Importance: This is indeed an essential part, which was unjustifiably left out of the first submission, originally to make it more suitable for a non-technical audience. Since the message derived from the feature importance is intuitively understandable and crucial for the explanation of the competition between TFs and nucleosomes, we added a part to the end of the results, in which we exhibit the best predictive input features to determine nucleosomal DNA. We identified a high meaning of HelT signals, especially in a long-range, low-frequency context. This is the according results section in the manuscript:

The random forest was selected for modeling the classifier underlying the NF score due to its intrinsic ability to generate feature importance scores. The input features consist of the power spectral density (PSD) of frequencies inherent in underlying DNASHape structures (see the Classifier section in Material and Methods). In Table 1, the top ten input features for distinguishing between nucleosomal and linker DNA are listed. The mean decrease in Gini impurity is given for all frequencies of the PSD transformed DNASHape values. It is evident that most of the best input features to distinguish between nucleosomal and linker DNA are low-frequency differences in HelT. In the original DNA structure space, low frequencies indicate long-range structural changes rather than short local perturbations.

Table 1: Feature importance of the top input features for the random forest classifier as mean decrease in Gini impurity. It can be seen that the top input features consist of low-frequency signals of HelT.

	Feature	Freq 1/147 bp	Gini Importance
1	HelT	5	62.16
2	HelT	1	60.33
3	HelT	2	55.22
4	HelT	4	52.84
5	HelT	3	47.88
6	HelT	6	40.64
7	ProT	4	34.45
8	ProT	3	27.75
9	ProT	5	25.33
10	ProT	8	23.84

As HelT denotes the rotational angle between two adjacent bp along the helix direction, this effect can be understood as what happens when taking both ends of a 147 bp DNA double strand and twisting them in opposite directions. Supplementary Fig S7 shows a visualization of the mean profiles for each DNASHape feature as well as the PSD transformed information for both nucleosomal and linker DNA from the training set and underlines the differences between both groups in the low-frequency structure changes visually. It has been shown in this study that the NF score is highest at regulatory meaningful positions which are targeted by TFs. It is worth noting that the long-range structural disparities between nucleosomal and linker DNA contrast with the localized and short DNA structure features known to be recognized by TFs at their binding sites [52]. Therefore, attraction for the binding of nucleosomes and for TFs is encoded at the same locations but on different frequency ranges.

This part is picked up in the Discussion at this position:

[...] Therefore, we interpret the crucial function of DNA in these "regulatory" nucleosomes as a mechanism to regulate the competition between nucleosomes and TFs for spatial availability. This competition affects the energy required to activate biological processes, allowing for dynamic control and modulation. This co-localized competition raises the question of how sequence-encoded nucleosome preferences and well-defined TFBSs can evolve at the same positions. The answer lies in the analysis of feature importance derived from the random forest model. This study demonstrates that the primary predictors for distinguishing between nucleosomal and linker DNA stem from low-frequency, i.e., long-range, differences in HelT. Simultaneously, it is well-documented that TFs recognize very local, high-resolution, i.e., high-frequency, DNA structure signals [52]. This leads to the assumption that the nucleosome attraction and the counteracting TFs are encoded at the same positions but on different frequencies of DNA structure changes. Further evidence [...]

Comparison to other models: It is correct that the model has a lower AUC than the one achieved with the SVM of Guo et al. To add a bit more focus on the classifier performance, we extended our performance measures of the PSD based classifier and included it as Table 2 in the main manuscript. Concerning the comparison with other tools we only mentioned the performance of the currently best performing tools on the same data to show the upper bounds of the possibilities and to show that our model does not exceed the performance of CNNs. That the accuracy shown here is slightly higher (about half a percent) than in the previous manuscript can be explained with a new evaluation run and falls within the standard deviation shown in the table.

Table 2: Performance measures for the random forest classifier, evaluated with 10-fold cross-validation. The numbers in parentheses denote the standard deviation of these measures within the 10 folds.

	Performance
Accuracy	82.01 % (± 1.55)
Sensitivity	85.36 % (± 2.29)
Specificity	78.93 % (± 3.26)
F1	85.10 % (± 1.88)
AUC	0.89 (± 0.01)
MCC	0.64 (± 0.03)

The performance of the random forest classifier based on 10-fold cross-validation is shown in Table 2. The accuracy is measured with 82.01 % (± 1.55) and the AUC with 0.89 (± 0.01). Furthermore, we compiled a range of different random forest models based on all of the previously mentioned data pre-processing steps. Supplementary Table S5 lists performance measures for classifiers trained on raw sequences, pure DNashape values and the final PSD frequency transformation. It demonstrates that with each further transformation, there is a substantial gain in performance, which underlines the suitability of the frequency modeling to cover the underlying periodicity proposed by Segal et al . [20].

The most recent classifiers use convolutional neural networks (CNNs) and reach accuracies of up to 0.889 (CoreNUP [75]) and an AUC of up to 0.94 (LeNUP [35]) on the given dataset. However, although CNNs are undoubtedly valuable for such prediction tasks in principle, we opted not to use them on this specific dataset due to their propensity for overfitting on smaller datasets. Furthermore, the random forest methodology enables us intrinsically to

derive feature importance scores from the model and we have a direct connection between the NF score resource and what determines the score on the base of periodic DNA structure changes in the form of PSD. The feature importance was obtained using the importance function from the randomForest package in R and is provided as mean decrease in Gini impurity. The ranking of each input feature by importance can be viewed in Table 1.

To evaluate these differences further, we internally did a cross-comparison with architectures based on different models employing convolutional neural networks. With a combination of convolutional and LSTM layers, as suggested by the CoreNUP model, we were able to increase the accuracy of the classifier on the same dataset by about 3 %. However, we decided to keep the random forest model due to the direct link between feature importance scores and the predictions we used for our analysis of nucleosome positioning in transcription with the NF score. In our opinion, the gain in absolute performance does not outweigh the advantage of explainability, especially when applied to biologically meaningful examples, which showed no significant differences within our internal studies. As described in the previous points, we added a new section about using the feature importance scores for explaining the relevant sequence features.

Baseline Models: We compared different transformations of the original input data to ensure the suitability of the transformations of DNashape and the PSD frequency transformation. The simplest way of modeling is as raw sequences, encoded with One-Hot-Encoding. The next step of complexity in our analysis are DNashape values. These are derived from k-mer lookup tables in the DNashape tool and serve as a baseline model similar to the ones proposed in the comment. Finally we add the frequency transformation step with the PSD. We include the classifier performance of models trained on each of these levels in Supplementary table S5, in which we show the gain in performance for each transformation step.

Supplementary Table S5. Performance for different stages of training data pre-processing, evaluated with 10-fold cross-validation. The numbers in parentheses denote the standard deviation of these measures within the 10 folds. Each transformation step exhibits a gain in performance

	Raw Sequence	DNashape	PSD Spectrum
Accuracy	71.24 % (± 1.88)	78.38 % (± 1.76)	82.01 % (± 1.55)
Sensitivity	72.12 % (± 4.16)	78.38 % (± 3.35)	85.36 % (± 2.29)
Specificity	70.58 % (± 3.41)	78.42 % (± 1.28)	78.93 % (± 3.26)
F1	69.70 % (± 2.40)	78.45 % (± 1.91)	85.10 % (± 1.88)
AUC	0.79 (± 0.03)	0.85 (± 0.01)	0.89 (± 0.01)
MCC	0.43 (± 0.05)	0.57 (± 0.03)	0.64 (± 0.03)

Other issues:

Given the overall concerns in the literature on the meaning of spatial separation in dimensionality reduction approaches such as UMAP and t-SNE, I am not sure Fig 1C and accompanying discussion adds much value, and would prefer to see more details on the random forest model in its place.

We agree that the t-SNE does not add a lot of information and we removed it along with the AUC visualization in the same Figure. Instead, we describe more details in the Classifier section about the backgrounds of dataset labeling, dimensionality of the input data for the random forest and the feature importance scores to explain the underlying sequence patterns to support nucleosome positioning. The corresponding parts in the manuscript are included in the replies to previous major comments already.

The conclusion that promoters are preferred in K562 and enhancers in GM12878, seems overreaching with no replicates.

This point was supposed to be merely descriptive based on the results in the figure. The corresponding part of the sentence is dropped, so that it is clear that we don't see any reason why there should be such a difference between K562 and GM12878 apart from simple experimental biases.

[...] This could be due to experimental biases but since the direction of the differences is not the same in both instances, ~~it appears that open promoters are generally favored in K562, while the enhancers are more accessible overall in GM12878.~~

The text and figure should describing the use of the GATA3 positive and negative MNase, should be made more clear to indicate which cells are measured and that the cells are +GATA3 and -GATA3. Some of this text is present in the methods, but it should be made more explicit in the main text.

This is a very helpful suggestion and we use the suggested labeling of both groups in the figure as well throughout the main text.

CTCF peaks and its array are not strand agnostic. CTCF binds to one or other strand, and the nucleosome array has been shown to be slightly asymmetric around the strand of the CTCF motif. Given that this asymmetry is minor, it sentence on the strand of peaks can be dropped.

The sentence is now dropped from the manuscript.

[...] The nucleosome occupancy and sequence-intrinsic nucleosome support around 43247 ChIP-seq peaks for CTCF in K562 and 43631 peaks in GM12878 are compared in detail in Fig 4B. ~~The peaks are agnostic of strand orientation, since they are not mapped to a particular gene.~~

The methods on how public datasets were used is insufficient. For the MNase datasets used, it should be made clear how the data reflects nucleosomal positioning. le single-end, paired-end, how is it smoothed, units etc. It isn't obvious how to interpret the given ref (44), if this is a URL please supply that.

The description of the dataset is indeed insufficient without an accessible link to the original protocols. There are URLs for the protocol descriptions on ENCODE which we now included in the References accordingly.

- [63] M. Snyder *et al.*, "General Protocol - Track description for UCSC Genome Browser composite track hg19/wgEncodeSydhNsome." 2011. Accessed: Sep. 01, 2023. [Online]. Available: <https://www.encodeproject.org/documents/912dc886-d576-48ac-aa75-58ddaee6d9ff/@download/attachment/wgEncodeSydhNsome.html.pdf>
- [64] A. Boyle *et al.*, "General Protocol - Track description for UCSC Genome Browser composite track hg19/wgEncodeOpenChromDnase." 2011. Accessed: Sep. 01, 2023. [Online]. Available: <https://www.encodeproject.org/documents/6878f54f-b81e-4064-a6ae-f78c76ed7f85/@download/attachment/wgEncodeOpenChromDnase.release3.html.pdf>
- [65] K. Onate, "Pipeline Protocol - ChIP pipeline: Details for both histone and transcription factor." 2016. Accessed: Sep. 01, 2023. [Online]. Available: https://www.encodeproject.org/documents/7009beb8-340b-4e71-b9db-53bb020c7fe2/@download/attachment/ChIP-seq_pipeline_overview.pdf

The word "chapters" at the end of the methods appears to refer to something beyond the paper.

The word refers to the beginning of the results section. The word chapters is confusing indeed and was changed to refer to the results.

[...] This is especially evident in the **first section of the results** about the counteracting relationship of high sequence-intrinsic nucleosome support in regulatory important regions against the low actual nucleosome occupancy *in vivo* at these positions of nucleosome-TF competition.

The exon and polyA site results seem striking - can the authors confirm that they are excluding first exons (ie TSS) from their exon starts results. Given the point about about short exons, the authors should either stratify their exon mid summary by exon size, or refine which exons are included to ensure the signal is distinct from exon starts.

We thank the reviewer for the advice concerning the first exons. This is indeed a relevant distinction for evaluating the message about sequence-supported nucleosome positioning over exons as a general statement. The figure contains all exons, without further discrimination. In order to understand the difference between first exons and all others, we now provide Supplementary Figure S5, which compares the nucleosome occupancy and NF score, separated by all exons except the first one and plots for all isolated first exons. This is a screenshot of the new figure:

Supplementary Figure S5. First Exons. MNase-seq and NF scores for all human exons except the first (top) and for only the first exons, essentially the TSS (bottom). It is evident that the exons excluding all first exons are characterized by nucleosome occupancy over their midpoints while first exons show only a clearly defined downstream nucleosome which can be interpreted as the +1 nucleosome.

It is evident that the overall message from the main text is consistent with the image shown here, in which the first exons have been filtered out due to the relative minority of first exons in the whole dataset. It is evident, however, that the distinction is meaningful since the nucleosome occupation over exon mids is not evident for the isolated first exons and instead they show the expected depletion of the TSS figure in the main text with the well-positioned +1 nucleosome downstream.

This effect is referenced in the manuscript with this sentence at the end of the exon section:

[...] The images shown here include all human exons. Since the first exon is essentially the TSS, Supplementary Fig S5 shows the same profiles, once for all exons except the first of each gene and once only for the first exons. Due to their comparatively small amount, the principal image does not change when filtering out the first exons. The isolated first exons, however, do not exhibit the same occupation of exons but rather show the same

nucleosome positioning as shown in the TSS analysis in Fig 5A with the clear +1 nucleosome.

A refined analysis with different exon sizes taken into account can be viewed in Supplementary Figures S3 and S4.

March 5, 2024

Re: Life Science Alliance manuscript #LSA-2023-02380-TR

Mr. Malte Sahrhage
University Medical Center Goettingen
Department of Medical Bioinformatics
Goldschmidtstr 1
Göttingen 37077
Germany

Dear Dr. Sahrhage,

Thank you for submitting your revised manuscript entitled "The importance of DNA sequence for nucleosome positioning in transcriptional regulation" to Life Science Alliance. The manuscript has been seen by an original reviewer whose comments are appended below. While the reviewer continues to be overall positive about the work in terms of its suitability for Life Science Alliance, some important issues remain.

Our general policy is that papers are considered through only one revision cycle; however, given that the suggested changes are relatively minor, we are open to one additional short round of revision. Please note that I will expect to make a final decision without additional reviewer input upon re-submission.

Please submit the final revision within one month, along with a letter that includes a point by point response to the remaining reviewer comments.

To upload the revised version of your manuscript, please log in to your account: <https://lsa.msubmit.net/cgi-bin/main.plex>
You will be guided to complete the submission of your revised manuscript and to fill in all necessary information.

B. MANUSCRIPT ORGANIZATION AND FORMATTING:

Sincerely,

Reviewer #2 (Comments to the Authors (Required)):

The authors have given an extensive rebuttal, with some nice additions including better insight into what the random forest classifier. Nevertheless, my principal concern about the distribution of the nucleosomal and non-nucleosomal sequences across the genome has not been addressed.

To make the point clear, the classifier is built on a subset of sequences that have well positioned nucleosomes in CD4+ cells and a set of sequences of less clear origin that have low probability of nucleosomes. It is therefore possible, that the positive set is over represented in TSS and enhancers, and the negative set is overrepresented in regions relatively inaccessible to MNase or regions without coherent nucleosomal positioning over multiple cells which will appear as low regions in a relative normalised MNase-seq. Therefore, their classifier may be capturing the differences in sequence features of these regions as opposed to nucleosome preference per se.

Therefore, a reasonable control for their method which would make it as transparent as possible, is to map their sequences to the genome to obtain the coordinates of all sequences and then provide a supplemental figure showing the proportion of regions at features such as TSS, gene body, exon, intron, enhancer, or even HMM states as originally suggested is necessary. The authors could further provide a supplemental bed file with coordinates.

If the authors find a substantial asymmetry between the positive and negative sets they do not need to change their conclusions, but state the implied limitation and that future work may be improved by more balanced sets of nucleosome positive and negative sequences. Should this analysis already exist elsewhere in literature using these datasets, the authors can of course cite this here instead.

Final other comment, the authors should explicitly state in the methods the total number of features input to their random forest classifier. As it is the full PSDs, presumably its $4 \times 147 = 588$ features? Although as these are smoothed PSDs some of these features are redundant, the authors need not mention the redundancy but should give the total number.

Response to Reviewers

Reviewer #2

The authors have given an extensive rebuttal, with some nice additions including better insight into what the random forest classifier. Nevertheless, my principal concern about the distribution of the nucleosomal and non-nucleosomal sequences across the genome has not been addressed.

To make the point clear, the classifier is built on a subset of sequences that have well positioned nucleosomes in CD4+ cells and a set of sequences of less clear origin that have low probability of nucleosomes. It is therefore possible, that the positive set is over represented in TSS and enhancers, and the negative set is overrepresented in regions relatively inaccessible to MNase or regions without coherent nucleosomal positioning over multiple cells which will appear as low regions in a relative normalised MNase-seq. Therefore, their classifier may be capturing the differences in sequence features of these regions as opposed to nucleosome preference per se.

Therefore, a reasonable control for their method which would make it as transparent as possible, is to map their sequences to the genome to obtain the coordinates of all sequences and then provide a supplemental figure showing the proportion of regions at features such as TSS, gene body, exon, intron, enhancer, or even HMM states as originally suggested is necessary. The authors could further provide a supplemental bed file with coordinates.

If the authors find a substantial asymmetry between the positive and negative sets they do not need to change their conclusions, but state the implied limitation and that future work may be improved by more balanced sets of nucleosome positive and negative sequences. Should this analysis already exist elsewhere in literature using these datasets, the authors can of course cite this here instead.

We thank the reviewer to clarify this point again and the editor for the chance to include the analysis in our response. As suggested, we mapped the training sequences used in this study to the human genome and calculated their distribution over different biological contexts, separated by nucleosomal or linker category. The result is included as **Supplementary Fig S10**:

It is evident in the image that there is no principal experimental bias that leads to an over-representation of sequences in regulatory important regions in nucleosomal sequences. Both categories are rather characterised by a frequency distribution that represents the general setup of the human genome with distal intergenic and intronic locations contributing the biggest part.

We added a sentence in the **Discussion** to reference the image in the manuscript:

[...] Conversely, there are instances where lower nucleosome support indicates non-functionality, as observed in constitutive heterochromatin. **That sequence-positioned nucleosomes in regulatory regions are a functional effect and not the result of experimental biases in relation to linker sequences in the underlying training data from Guo et al. is illustrated by Supplementary Fig S10. This image compares the training sequences in terms of their relative proportion over different biological contexts. There are no major differences in the origin of both categories, which are mainly distributed over the most abundant regions of the human genome such as distal intergenic regions and introns.**

[...]

Additionally, we include the BED files that were produced to address this point in the Supplementary Information and reference them in the **Classifier** section:

[...] It contains 4573 non-redundant FASTA sequences which are categorized as either nucleosomal (n=2273) or linker sequences (n=2300). **A mapping of these sequences to the hg19 human reference genome is additionally provided as BED files (See Data Availability).** The labeling strategy [...]

...as well as in the **Data Availability** section:

[...]

The Guo et al. training dataset was further mapped to the human reference genome hg19 in the form of BED files using the Rsubread R package [77]. These mappings are appended as additional Supplementary Files SD1 and SD2, separated by nucleosomal and linker sequences.

Final other comment, the authors should explicitly state in the methods the total number of features input to their random forest classifier. As it is the full PSDs, presumably its $4 \times 147 = 588$ features? Although as these are smoothed PSDs some of these features are redundant, the authors need not mention the redundancy but should give the total number.

We agree that the specification of feature dimensions is beneficial to include in the manuscript. We were aware of the redundancy of the resulting PSD. Therefore, we exclude the redundant features and use only half of the PSD for training input. The feature space is further reduced slightly due to the k-mer lookup nature of DNashape. The outer edges of the sequences where the k-mer mask extends over the edges by a few nucleotides cannot be calculated correctly and are therefore left out. We added a respective part in the description of the **Classifier** setup, where the detailed setup to reach the final 288 input features is explained.:

[...] The result of the classification is the categorization of a sequence being either *nucleosomal* or *linker* and serves as the basis for the two types of NF scores that will be described further in the NF score section. **The particular final number of input features at the**

end of the transformation is 288. This results from the basic sequence lengths (147 bp), the number of DNASHape parameters used for training (4) and a subtraction of some missing DNASHape values at the edges of the sequence where the centered k-mer mask is incomplete (12). These 576 features are halved due to the redundant nature of both sides of a full PSD spectrum to the final number of 288 input features.

May 9, 2024

RE: Life Science Alliance Manuscript #LSA-2023-02380-TRR

Malte Sahrhage
University Medical Center Goettingen
Department of Medical Bioinformatics
Goldschmidtstr 1
Göttingen 37077
Germany

Dear Dr. Sahrhage,

Thank you for submitting your revised manuscript entitled "The importance of DNA sequence for nucleosome positioning in transcriptional regulation". We would be happy to publish your paper in Life Science Alliance pending final revisions necessary to meet our formatting guidelines.

- please be sure that the authorship listing and order is correct
- please upload your main and supplementary figures as single files
- please upload your Tables in editable .doc or Excel format
- please add your main, supplementary figure, and table legends to the main manuscript text after the references section
- please incorporate the Supplementary References into the main Reference list
- please add the Twitter handle of your host institute/organization as well as your own or/and one of the authors in our system
- please remove the Graphical Abstract from the manuscript file and leave it uploaded separately
- please remove your figures from the manuscript file
- please add callouts for Figure 6A-B to your main manuscript text

A. FINAL FILES:

B. MANUSCRIPT ORGANIZATION AND FORMATTING:

Sincerely,

May 16, 2024

RE: Life Science Alliance Manuscript #LSA-2023-02380-TRRR

Mr. Malte Sahrhage
University Medical Center Goettingen
Department of Medical Bioinformatics
Goldschmidtstr 1
Göttingen 37077
Germany

Dear Dr. Sahrhage,

Thank you for submitting your Research Article entitled "The importance of DNA sequence for nucleosome positioning in transcriptional regulation". It is a pleasure to let you know that your manuscript is now accepted for publication in Life Science Alliance. Congratulations on this interesting work.

DISTRIBUTION OF MATERIALS:

Again, congratulations on a very nice paper. I hope you found the review process to be constructive and are pleased with how the manuscript was handled editorially. We look forward to future exciting submissions from your lab.

Sincerely,
